# Electron density modulation of $NiCo_2S_4$ nanowires by nitrogen incorporation for highly efficient hydrogen evolution catalysis

Yishang Wu[1,2], Xiaojing Liu[1], Dongdong Han[1], Xianyin Song[3], Lei Shi[1], Yao Song[1,2], Shuwen Niu[1], Yufang Xie[1], Jinyan Cai[1], Shaoyang Wu[1], Jian Kang[1], Jianbin Zhou[1], Zhiyan Chen[2], Xusheng Zheng[4], Xiangheng Xiao [3] & Gongming Wang [1]

Metal sulfides for hydrogen evolution catalysis typically suffer from unfavorable hydrogen desorption properties due to the strong interaction between the adsorbed H and the intensely electronegative sulfur. Here, we demonstrate a general strategy to improve the hydrogen evolution catalysis of metal sulfides by modulating the surface electron densities. The N modulated $NiCo_2S_4$ nanowire arrays exhibit an overpotential of 41 mV at 10 mA cm$^{-2}$ and a Tafel slope of 37 mV dec$^{-1}$, which are very close to the performance of the benchmark Pt/C in alkaline condition. X-ray photoelectron spectroscopy, synchrotron-based X-ray absorption spectroscopy, and density functional theory studies consistently confirm the surface electron densities of $NiCo_2S_4$ have been effectively manipulated by N doping. The capability to modulate the electron densities of the catalytic sites could provide valuable insights for the rational design of highly efficient catalysts for hydrogen evolution and beyond.

[1] Department of Chemistry, University of Science and Technology of China, Hefei, Anhui 230026, China. [2] School of Materials Science and Engineering, Central South University of Forestry and Technology, Changsha 410004, China. [3] Department of Physics, Wuhan University, Wuhan 430072, China. [4] National Synchrotron Radiation Laboratory, University of Science and Technology of China, Hefei 230029, China. These authors contributed equally: Yishang Wu, Xiaojing Liu  Correspondence and requests for materials should be addressed to X.Z. (email: xxh@whu.edu.cn) or to X.X. (email: zxs@ustc.edu.cn) or to wanggm@ustc.edu.cn)

With the ever-increasing consumption of fossil fuels and the associated environmental pollution, hydrogen with high gravimetric energy density has been considered as a promising energy carrier due to its renewability, cleanness, and carbon neutrality[1–4]. Water electrolysis, as one of the most feasible methods for scalable hydrogen production, still suffers from low efficiency and highly expensive catalysts[5,6]. To date, platinum (Pt) is still the most efficient hydrogen evolution reaction (HER) catalyst, while the high cost and low crustal abundance greatly impede its extensive applications[7]. To this end, transition metal compounds such as metal sulfides[8–13], selenides[14,15], phosphides[16–18], nitrides[19,20], carbides[21,22], and metal alloys[23,24] have been widely studied as low-cost and reasonable alternatives to the precious Pt-based catalysts. Besides, inspired by the structures and compositions of the nitrogenase and hydrogenase in natural biological systems, the transition metal sulfides have attracted special interest as HER catalysts[13]. For example, layered $MoS_2$ was identified to be highly active toward HER with an overpotential of 187 mV at 10 mA m$^{-2}$ in acidic condition[8]. The cubic pyrite-phase $CoS_2$ with metallic characteristics and superior chemical stability exhibited an overpotential of 145 mV at the current density of 10 mA cm$^{-2}$ for HER[9]. Furthermore, the HER activities of the metal sulfides can be further enhanced through the introduction of another promoter species to form binary metal sulfides[25]. For instance, incorporation of Co into $MoS_2$ could efficiently promote the HER activity of $MoS_2$, achieving 3.5 mA cm$^{-2}$ at the overpotential of 100 mV[26]. In addition, Wang et al.[12] reported that Co-doped $FeS_2$ nanosheets-carbon nanotubes as highly active and stable catalysts for HER display a low overpotential of ~120 mV at 20 mA cm$^{-2}$ in acidic condition. Although the obtained efficiencies of metal sulfides are still far less than the benchmark Pt catalysts, the current studies indicate that transition metal sulfides with wide assortments of materials and tunable surface properties hold great promise for the development of highly efficient HER catalysts.

$NiCo_2S_4$ with excellent electrical conductivity, hybrid d orbitals and versatile redox nature, has displayed excellent catalytic performance for oxygen evolution and oxygen reduction reactions[27,28]. Very recently, $NiCo_2S_4$ was also employed for HER catalysis, delivering an overpotential of 210 mV at 10 mA cm$^{-2}$ in alkaline conditions[29]. The unsatisfactory catalytic activity is commonly attributed to the strong interaction between the adsorbed H (H*) and the strongly electronegative sulfur sites, which substantially hinders the H* from desorbing to produce free $H_2$[12,26,30]. Moreover, given that the strong electronegativity is the essential feature of S, the unfavorable H* desorption behavior is also a common drawback for most metal sulfides for HER catalysis. Although morphological engineering can improve the catalytic performance of metal sulfides to some extent by increasing the surface area, it cannot fundamentally alter the intrinsic catalytic properties of metal sulfides. In contrast, surface engineering raises new opportunities to manipulate the electronic properties of materials; however, it is also very difficult to directly weaken the strong H–S interaction. Therefore, pushing the HER activities of metal sulfides to the level of Pt catalysts by rationally modulating the electronic properties and unraveling the essential modulation mechanism is desirable but challenging.

Here we demonstrate a facile and general strategy to modulate the electron densities of the catalytic sites of $NiCo_2S_4$ nanowire arrays (NWs) for high performance HER activity by using nitrogen as the modulator. Instead of directly weakening the H–S interaction, we sacrifice part of the catalytic sulfur sites by replacing them with N, and indirectly facilitate the H* desorption from the remaining sulfur sites. The design principle is to decrease the electron densities of the sulfur sites by employing the strong interaction between metal and nitrogen. The optimized N-modulated $NiCo_2S_4$ (N–$NiCo_2S_4$) NWs achieve an overpotential of 41 mV at the current density of 10 mA cm$^{-2}$ and a Tafel slope of 37 mV dec$^{-1}$ for HER catalysis, which are very close to the performance of commercial Pt/C catalysts in alkaline conditions and also represent the best HER activity among the metal sulfides, to the best of our knowledge. X-ray photoelectron spectroscopy (XPS), synchrotron-based X-ray absorption near-edge structure (XANES) spectroscopy, ultraviolet photoemission spectroscopy (UPS), extended X-ray absorption fine structure (EXAFS), and density functional theory (DFT) studies consistently confirm the surface electron densities and d-band centers of $NiCo_2S_4$ can be effectively manipulated by N doping. To further unravel the effects of N on the catalytic process, DFT simulation reveals that the N incorporation could not only facilitate the desorption of H* from S sites but also assist the dissociation of $H_2O$, making it preferential to proceed on a lower potential energy surface. More importantly, we also prove that the N modulation is a general strategy to consistently enhance the HER catalysis of various metal sulfides including $NiCo_2S_4$, $FeS_2$, $CoS_2$, and $NiS_2$. The capability to modulate the electron densities and d band centers of the catalytic sites by rational surface engineering could provide valuable insights for the development of highly efficient HER catalysts and beyond.

## Results

**Synthesis and structural characterization**. N–$NiCo_2S_4$ nanowire arrays were synthesized via a two-step process, as shown by the schematic illustration in Fig. 1. Ni foam was selected as the growth substrate due to its excellent electrical conductivity and three-dimensional (3D) porous structure for fast ion diffusion. First, Ni-Co-carbonate hydroxides (Ni–Co–O) NWs were directly grown on the Ni foam through a well-developed hydrothermal synthesis, with $Co(NO_3)_2$ and $Ni(NO_3)_2$ as the metal sources and urea as the pH regulator (see Methods for details). The in-situ growth could endow the prepared electrodes with an intimate contact between the Ni–Co–O NWs and Ni foam. Then, N–$NiCo_2S_4$ NWs were prepared by co-sulfurization and nitridation of the Ni–Co–O NWs in a home-built tube furnace system with sulfur powder as sulfur source and ammonia gas as nitrogen source and carrier gas. As a control sample, $NiCo_2S_4$ without nitrogen doping was achieved by sulfurization of Ni–Co–O with argon as the carrier gas.

The morphological and structural analyses were further performed using scanning electron microscopy (SEM), transmission electron microscopy (TEM) and X-ray diffraction (XRD). Figure 2a and Supplementary Fig. 1a display the SEM images of the as-prepared Ni–Co–O NWs on Ni foam. Clearly, the Ni foam is uniformly covered with quasi-vertically aligned nanowires, with smooth surfaces and an average diameter of ~100 nm. The XRD pattern of the prepared Ni–Co–O NWs in Supplementary Fig. 2

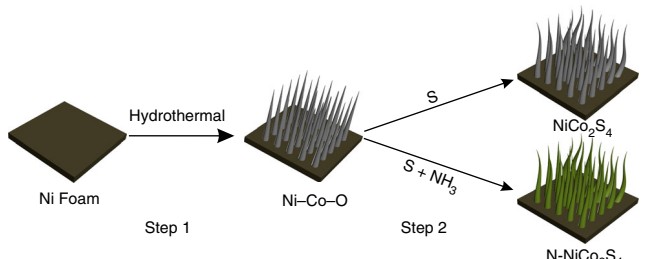

**Fig. 1** Schematic illustration of the fabrication of $NiCo_2S_4$ and N–$NiCo_2S_4$ nanowires through a two-step synthesis. Step 1: hydrothermal synthesis of Ni–Co–O nanowire arrays. Step 2: sulfurization and nitridation of Ni–Co–O to prepare $NiCo_2S_4$ and N–$NiCo_2S_4$ nanowire arrays

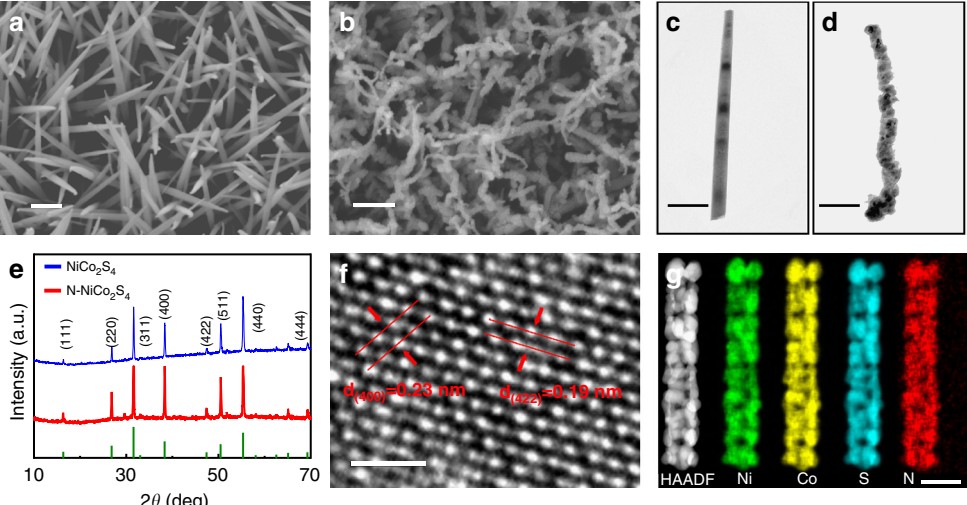

**Fig. 2** Morphological and structural characterization. SEM images for **a** Ni–Co–O and **b** N–NiCo$_2$S$_4$ nanowire arrays. Scale bar=400 nm. TEM images of **c** Ni–Co–O and **d** N–NiCo$_2$S$_4$ nanowires. Scale bar=200 nm. **e** XRD patterns of NiCo$_2$S$_4$ and N–NiCo$_2$S$_4$. **f** HRTEM image of N–NiCo$_2$S$_4$ nanowire array. Scale bar=1 nm. **g** HAADF-STEM image of a single N–NiCo$_2$S$_4$ nanowire and element mapping images of Ni, Co, S, and N elements in the N–NiCo$_2$S$_4$ nanowire. Scale bar=100 nm

can be indexed to Ni-Co-carbonate hydroxides[29]. After sulfurization and/or nitridation at 600 °C, the SEM images in Fig. 2b and Supplementary Fig. 1b–d reveal that the nanowire morphology is maintained but exhibits a rough surface, which can also be further confirmed by the TEM images of the nanowires before and after co-sulfurization and nitridation (Fig. 2c, d). With the increase of the annealing temperatures from 400 to 700 °C, the NWs become rougher, and finally aggregate into large particles at 700 °C (Supplementary Fig. 3). Figure 2e shows the XRD patterns of NiCo$_2$S$_4$ and N–NiCo$_2$S$_4$ exhibit very similar profiles, where the characteristic peaks located at 16.3°, 26.8°, 31.6°, 38.3°, 47.4°, 50.5°, and 55.3° 2$\theta$ can be well assigned to the (111), (220), (311), (400), (422), (511), and (440) planes of cubic NiCo$_2$S$_4$ phase (JCPDS Card No. 20-0782), respectively[27,29]. Besides, no other discernible impurity peaks are detected after nitrogen doping, indicating that nitrogen doping does not change the structural phase of NiCo$_2$S$_4$. Moreover, the well resolved lattice fringes of the high-resolution TEM (HRTEM) image of N–NiCo$_2$S$_4$ nanowire with interplanar distances of 0.23 nm and 0.19 nm can be assigned to the (400) and (422) planes of the NiCo$_2$S$_4$, further revealing the structural phase of NiCo$_2$S$_4$ is still maintained after nitrogen doping (Fig. 2f). In addition, the high angle annular dark-field scanning TEM (HAADF-STEM) and element mapping images in Fig. 2g provide clear visual evidences that Co, Ni, S, and N elements are homogeneously distributed on the whole N–NiCo$_2$S$_4$ nanowire. Moreover, the element mapping images with higher magnification also display uniform element distribution of N and S in N–NiCo$_2$S$_4$ (Supplementary Fig. 4), suggesting that nitrogen is uniformly doped into the NiCo$_2$S$_4$ nanowire.

**Characterization of the chemical states**. To elucidate the effects of the nitrogen dopants on the chemical states of NiCo$_2$S$_4$, X-ray photoelectron spectroscopy (XPS) and synchrotron-based X-ray absorption near-edge structure (XANES) spectroscopy were conducted. XPS survey spectrum collected on N-NiCo$_2$S$_4$ reveals the presence of Ni, Co, S, and N elements (Supplementary Fig. 5 and 6), consistent with the element mapping images in Fig. 2g and Supplementary Fig. 4. The carbon signal is believed to be involved during sample preparation, which is used for the calibration of the binding energy. The surface N/S ratio is determined

to be around ~0.12, based on the XPS results. Figure 3a shows the XPS Co 2p core-level spectra of NiCo$_2$S$_4$ and N–NiCo$_2$S$_4$ NWs. Both spectra display one pair of peaks arising from the spin-orbit doublet of Co 2p, which can be assigned to Co 2p$_{3/2}$ and Co 2p$_{1/2}$. The Co 2p$_{3/2}$ and Co 2p$_{1/2}$ peaks of the pristine NiCo$_2$S$_4$ locate at 778.3 and 793.5 eV, consistent with the literature values of NiCo$_2$S$_4$, shown in Supplementary Fig. 6a[27]. Interestingly, the binding energies of Co 2p$_{3/2}$ and Co 2p$_{1/2}$ display a positive shift of ~0.4 eV after nitrogen incorporation. Meanwhile, similar shift of 0.5 eV to higher energy region with N doping is also observed in XANES studies (Fig. 3b). The Co L-edge spectra of NiCo$_2$S$_4$ and N-NiCo$_2$S$_4$ own two photo energy peaks of L$_3$ and L$_2$, located at 780.6 and 795.7 eV, corresponding to the electron transition from Co 2p$_{1/2}$ and 2p$_{3/2}$ to 3d, respectively[31]. Besides, the binding energies and photo energies of XPS Ni 2p and Ni L-edge spectra of NiCo$_2$S$_4$ also consistently present positive shifts after N incorporation (Fig. 3c, and Supplementary Fig. 6b,d). The positive shifts in XPS 2p and L-edge spectra of metal atoms can be attributed to the strong interaction between nitrogen and metal atoms, which is further proved by the presence of N-metal bonds in XPS N 1 s spectrum of N-NiCo$_2$S$_4$, as shown in Supplementary Fig. 6c[32]. Furthermore, EXAFS measurement also indicates the decreased coordination number of Co–S is attributed to the introduced N, which occupies the S sites to form Co–N (Supplementary Fig. 7). N, with even stronger electronegativity than S, can further withdraw electrons around metal atoms, resulting in the reduction of the electron density localization around the metal atoms and thus the observation of the positive shift in binding energies and photon energies of metal elements in XPS and XANES measurements.

On the other hand, XPS and XANES were also employed to investigate the effects of nitrogen on the chemical states of S atoms in NiCo$_2$S$_4$ and N–NiCo$_2$S$_4$. Figure 3e shows the XPS S 2p core-level spectra of NiCo$_2$S$_4$ and N–NiCo$_2$S$_4$, respectively. The broad peak of S 2p can be deconvoluted into a pair of peaks located at around 162.4 eV and 163.6 eV (Supplementary Fig. 6d), which can be assigned to S 2p$_{3/2}$ and S 2p$_{1/2}$, respectively[12,33]. The binding energy positions of S 2p in NiCo$_2$S$_4$ are also consistent with the reported values of conventional metal sulfides, suggesting the existence of metal-S bonds[9,34]. Interestingly, a small positive shift (0.3 eV) of S 2p XPS spectrum can be observed in

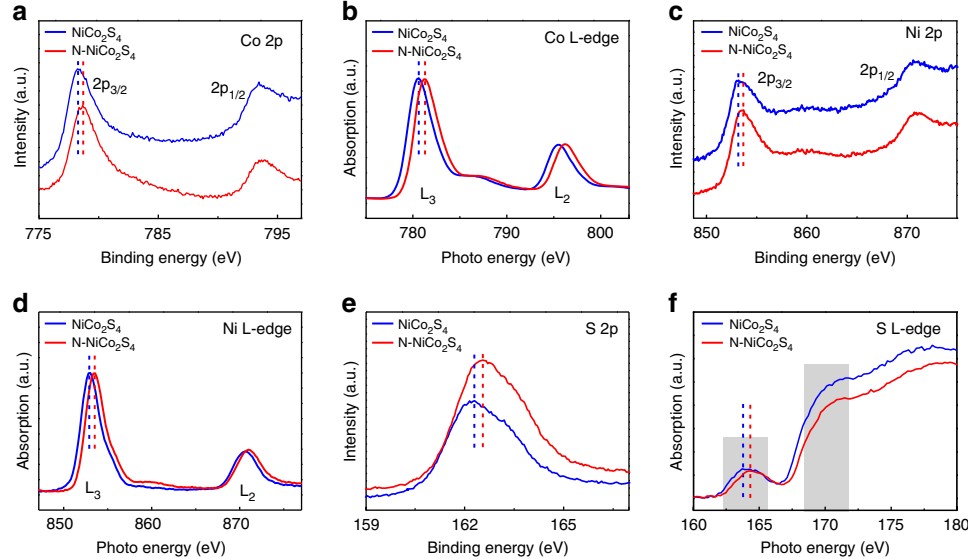

**Fig. 3** XPS and XANES characterization of $NiCo_2S_4$ and $N–NiCo_2S_4$. XPS core-level spectra of **a** Co 2p, **c** Ni 2p, and **e** S 2p for $NiCo_2S_4$ (blue) and $N–NiCo_2S_4$ (red). XANES spectra of **b** Co L-edge, **d** Ni L-edge, and **f** S L-edge for $NiCo_2S_4$ (blue) and $N–NiCo_2S_4$ (red)

$N–NiCo_2S_4$, suggesting the strong metal-N interaction could indirectly weaken the metal-S bonds by decreasing the electron density around S atoms. Furthermore, Fig. 3f displays the S L-edge spectra of $NiCo_2S_4$ and $N–NiCo_2S_4$, where the peaks at around 164 eV and 170 eV of S L-edge are assigned as the photoelectron transition from S 2p state to S 3 s σ* and the empty S 3d states[35]. As expected, the S L-edge spectrum of $N–NiCo_2S_4$ also exhibited a small positive shift (0.2 eV) in photo energy, further indicating N can work as an electron density modulator and indirectly manipulate the electron densities around S. It can be understood that the decreased electron densities around metal atoms induced by the strong metal-nitrogen bonds could further weaken the nearby metal-S bonds, which can well explain the positive shifts of the binding energy and photo energy in S 2p and S L-edge spectra. Furthermore, UPS measurement indicates the d band center shifts far from the Fermi level ($E_F$) after N doping (Supplementary Fig. 8), suggesting that metal-N interaction makes a significant contribution to valence band structure of $NiCo_2S_4$. Overall, all these results indicate N incorporation could effectively regulate the electron densities of $NiCo_2S_4$, with the assistance of the strong interaction between metal and nitrogen.

**Enhanced activity for alkaline hydrogen evolution**. The HER activity of $N–NiCo_2S_4$ was evaluated using a three-electrode setup in 1.0 M KOH aqueous solution, with the prepared material as the working electrode, Ag/AgCl (saturated KCl solution) as the reference electrode and graphite rod as the counter electrode, respectively (see Methods for details). Besides the $N–NiCo_2S_4$, the bare Ni foam, Ni–Co–O, $N–NiCoO$ and $NiCo_2S_4$ and commercial Pt/C (20 wt%) were also studied as control samples. Figure 4a presents the linear sweep voltammetry (LSV) measurements performed at a sweep rate of 5 mV s$^{-1}$. Clearly, the $N–NiCo_2S_4$ exhibits significantly improved current density and much decreased onset potential, indicating the HER catalytic performance of $N–NiCo_2S_4$ is substantially better than those of the bare Ni, Ni–Co–O, $N–NiCoO$ and $NiCo_2S_4$, and even close to the performance of the benchmark Pt/C. The temperature dependent HER activities of $N–NiCo_2S_4$ were also studied (Supplementary Fig. 9). With the increase of temperatures, the catalytic performance is gradually increased and then achieves a maximum activity at the temperature of 600 °C. As the temperature is

further increased above 600 °C, the performance decreases probably due to the aggregation of $N–NiCo_2S_4$ NWs into large particles (Supplementary Fig. 3). Even so, all the $N–NiCo_2S_4$ prepared at different temperatures still consistently display lower onset potentials and higher current densities for hydrogen evolution than the pristine $NiCo_2S_4$, suggesting nitrogen doping can intrinsically change the catalytic properties of $NiCo_2S_4$. In addition, the Tafel slope is another important parameter to evaluate the dominant reaction mechanism in the HER process. Figure 4b shows the corresponding Tafel plots of the studied electrodes. As expected, the Tafel slope and overpotential of $N–NiCo_2S_4$ is consistently smaller than those of the Ni foam, Ni–Co–O, $N–NiCoO$ and $NiCo_2S_4$, and also close to the commercial Pt/C. Figure 4c summarizes and compares the overpotentials at the current density of 10 mA cm$^{-2}$ and the Tafel slopes of the bare Ni foam, Ni–Co–O, $N–NiCoO$, $NiCo_2S_4$, $N–NiCo_2S_4$, and Pt/C. The overpotentials for the bare Ni foam, Ni–Co–O, $N–NiCoO$, $NiCo_2S_4$, $N–NiCo_2S_4$ and Pt/C at 10 mA cm$^{-2}$ are 213, 170, 124, 104, 41, and 28 mV, respectively. More importantly, the overpotential of 41 mV for $N–NiCo_2S_4$ is very close to the 28 mV of Pt/C at alkaline condition, exceeding the performance of the previously reported $NiCo_2S_4$ and, to the best of our knowledge, representing the best reported performance for the metal sulfides for HER so far, such as $MoS_2$, $WS_2$, $NiCo_2S_4$, $FeS_2$, $CoS_2$, and $NiS_2$[8–10,12,29,36,37], which is summarized in the Supplementary Table 1. Meanwhile, the derived Tafel slopes are 133, 153, 100, 78, 37, and 29 mV dec$^{-1}$ for the bare Ni foam, Ni–Co–O, $N–NiCoO$, $NiCo_2S_4$, $N–NiCo_2S_4$ and Pt/C, respectively. The slope of 37 mV dec$^{-1}$ for $N–NiCo_2S_4$ suggests the rate determining step could be the Heyrovsky process ($H_2O + M-H^*$ $+ e^- = M + H_2 + OH^-$) in alkaline condition, which means the rate-determining HER process of $N–NiCo_2S_4$ is closely related to water dissociation and $H^*$ desorption[38,39].

Considering defects such as partial oxidization or sulfur defects may be introduced during material preparation, we systematically study the effects of these defects on the HER catalysis and evaluate their contributions to the total performance. For the defects induced by the surface oxidation, we intentionally heated the $N–NiCo_2S_4$ in air under different temperatures to partially oxidize the surface of $N–NiCo_2S_4$. With increasing temperature, the HER performance continuously decreases, suggesting the surface oxidation has a negative effect on the HER catalysis and

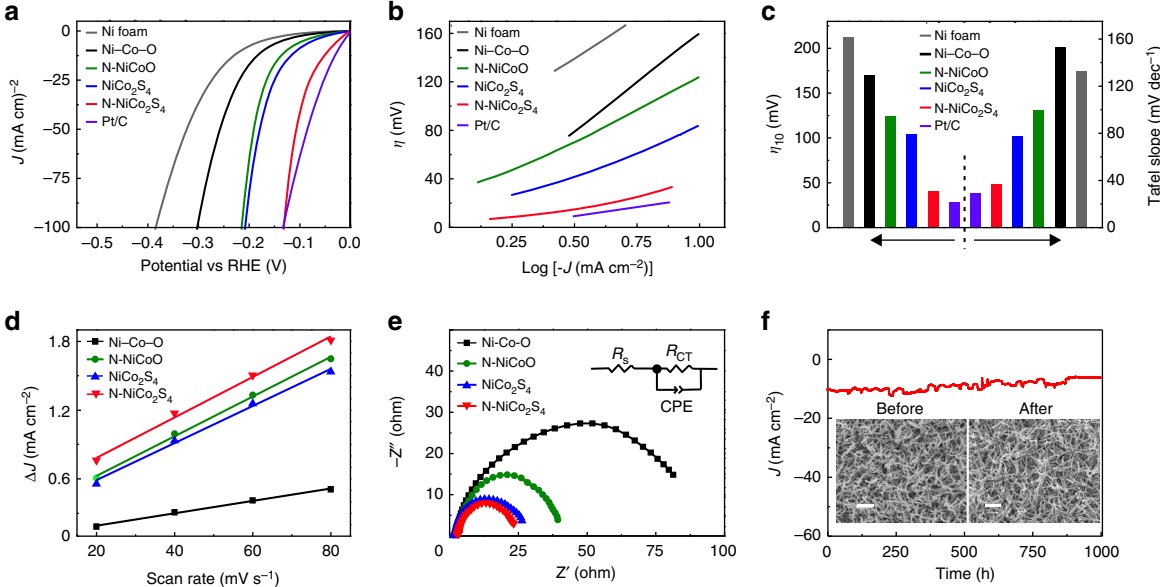

**Fig. 4** Electrochemical HER performance. **a** LSV curves with a scan rate of 5 mV s$^{-1}$ in 1.0 M KOH solution, **b** Tafel plots, and **c** the comparison of overpotential at 10 mA cm$^{-2}$ and Tafel slopes, for the bare Ni foam, Ni–Co-O, N-NiCoO, NiCo$_2$S$_4$, N-NiCo$_2$S$_4$ and Pt/C, respectively. **d** The plots of ΔJ versus scan rates for the Ni-Co-O, N-NiCoO, NiCo$_2$S$_4$, and N-NiCo$_2$S$_4$, respectively. **e** Nyquist plots of the Ni-Co-O, N-NiCoO, NiCo$_2$S$_4$ and N-NiCo$_2$S$_4$ at 50 mV vs. RHE. The inset is the equivalent circuit model that contains the electrolyte resistance ($R_s$), constant phase element (CPE) and charge-transfer resistance ($R_{CT}$). **f** Current density versus time (i–t) curves of N-NiCo$_2$S$_4$ NWs recorded for 1000 h at 50 mV vs. RHE, without IR correction. The SEM images for N-NiCo$_2$S$_4$ before and after stability test are illustrated as the insets; the scale bar is 1 μm

the observed high performance of N-NiCo$_2$S$_4$ in our work does not originate from the surface oxidation (Supplementary Fig. 10). Besides, the defects of sulfur vacancies were also introduced by annealing the NiCo$_2$S$_4$ in hydrogen conditions at 300 °C. Although the overpotential (81 mV at 10 mA cm$^{-2}$) of hydrogen-treated NiCo$_2$S$_4$ is smaller than that of pristine NiCo$_2$S$_4$ (Supplementary Fig. 11), the overall performance is still much less than that of N-NiCo$_2$S$_4$, which indicates that the sulfur vacancy has a positive effect, but is not the main contributor for the HER performance of N-NiCo$_2$S$_4$. Given that the variation of surface area after sulfurization and/or nitridation may also affect the catalytic performance, electrochemical cyclic voltammograms (CVs) was employed to estimate the relative surface areas. Typically, electrochemical double-layer capacitance ($C_{dl}$) is linearly proportional to the surface area of electrode and the $C_{dl}$ can be derived from the CV plots in the capacitive behavior region[11]. Supplementary Fig. 12 displays the CVs with various scan rates in the potential region of 0.1–0.2 V vs. reversible hydrogen electrode (RHE) and the rectangular shapes of CV curves indicate the electrochemical double-layer capacitive behavior. The $C_{dl}$ can be obtained by fitting the plots of the ΔJ ($j_a$–$j_c$) at 0.15 V against the scan rates (Fig. 4d), to estimate the electrochemical surface area (ECSA). The calculated capacitance and the relative surface area are summarized in Supplementary Table 2. The $C_{dl}$ of NiCo$_2$S$_4$ only slightly increases after N incorporation from 17 mF cm$^{-2}$ (NiCo$_2$S$_4$) to 18 mF cm$^{-2}$ (N-NiCo$_2$S$_4$), which means that N-NiCo$_2$S$_4$ NWs have almost the same surface area as NiCo$_2$S$_4$ NWs. Yet, the current density of N-NiCo$_2$S$_4$ was 4.5-fold higher than that of NiCo$_2$S$_4$ at 100 mV vs. RHE, revealing the enhancement does not originate from the surface area variation but the introduction of nitrogen which intrinsically changes the surface catalytic properties of NiCo$_2$S$_4$. The turnover frequency (TOF) is the most effective figure of merit to gain the insight into the intrinsic per-site activity of a catalyst. As illustrated in Supplementary Fig. 13, the TOF of N-NiCo$_2$S$_4$ is significantly higher than that of NiCo$_2$S$_4$, suggesting N dopant can intrinsically improve the catalytic activity of

NiCo$_2$S$_4$. In order to further elucidate the effects of N, we conducted the subsequent treatments to prepare N-NiCo$_2$S$_4$, instead of the aforementioned co-treatment with sulfurization and nitridation. Pre-doping N into Ni-Co-O followed by sulfurization, labeled as N-NiCo$_2$S$_4$ (N-S), and pre-sulfurization of Ni-Co-O followed by N doping, labeled as N-NiCo$_2$S$_4$ (S-N), were also prepared for HER catalysis. Interestingly, both of the XRD patterns (Supplementary Fig. 14a) are consistent with those of NiCo$_2$S$_4$ and N-NiCo$_2$S$_4$, suggesting the subsequent treatments also facilitates the formation of the NiCo$_2$S$_4$ phase. The HER catalytic performances of N-NiCo$_2$S$_4$ (N-S) and N-NiCo$_2$S$_4$ (S-N) are consistently higher than that of NiCo$_2$S$_4$, but lower than that of N-NiCo$_2$S$_4$ prepared by the co-treatment method (Supplementary Fig. 14b), probably due to the more uniform N distribution in N-NiCo$_2$S$_4$. Apparently, all these results firmly indicate that N plays an essential role in modulating the catalytic surface of NiCo$_2$S$_4$ and facilitating the hydrogen evolution reaction.

Furthermore, electrochemical impedance spectroscopy (EIS) measurements were conducted to investigate the N effects on the interfacial charge-transfer kinetics, which is also an important parameter to evaluate the intrinsic kinetic properties of electrocatalysts. Figure 4e shows the Nyquist plots of Ni-Co-O, N-NiCoO, NiCo$_2$S$_4$, and N-NiCo$_2$S$_4$, respectively. All the Nyquist plots display arc or semi-circles under the frequent region from 10 kHz to 0.1 Hz, which can be well fitted using a simple equivalent circuit composed of electrolyte resistance ($R_s$), constant phase element (CPE) and charge-transfer resistance ($R_{CT}$) (Fig. 4e, inset). The fitted $R_{CT}$ values for Ni-Co-O, N-NiCoO, NiCo$_2$S$_4$, and N-NiCo$_2$S$_4$ are 93, 61, 25, and 20 Ω, respectively. Importantly, the N-NiCo$_2$S$_4$ exhibits the smallest $R_{CT}$ value among the tested samples, suggesting the superior interfacial charge-transfer kinetics on the surface of N-NiCo$_2$S$_4$ for HER catalysis. Chronoamperometry was finally used to evaluate the long-term stability of the highly efficient N-NiCo$_2$S$_4$ under HER condition. Figure 4f displays the chronoamperometry curve at the potential of 50 mV vs. RHE in 1.0 M KOH solution

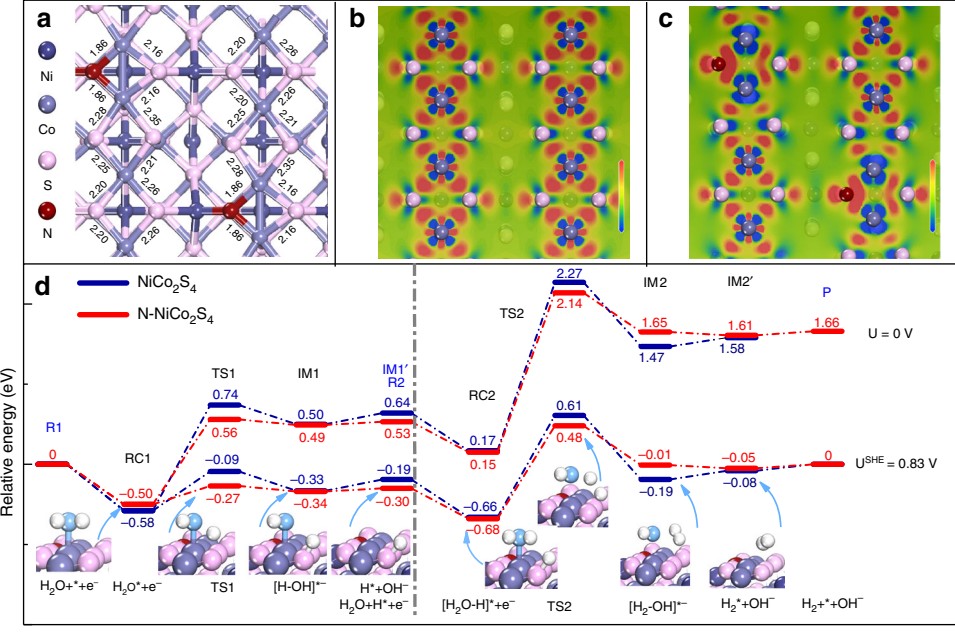

**Fig. 5** DFT calculations of $NiCo_2S_4$ and $N-NiCo_2S_4$. **a** The top-view structures of $N-NiCo_2S_4$ (100) with labeled bond lengths (Å) of Co–S and Co–N bonds. The top-view electron density difference of **b** $NiCo_2S_4$ (100), and **c** $N-NiCo_2S_4$ (100), ranging from −0.1 to 0.1. **d** Relative energy profiles and the simplified surface structures of the various reaction species along the reaction pathway, including the H* formation process (left panel) and $H_2$ formation process (right panel) in alkaline on the $NiCo_2S_4$ (100) and $N-NiCo_2S_4$ (100) at electrode potential $U = 0$ V and $U^{SHE} = 0.83$ V, respectively. R Reactant, RC Reactant Complex, TS Transition State, IM Intermediate, P Product

for a continuous 1000 h. We found that ~68.2% of the initial current density can be maintained after 1000 h testing, indicating its excellent electrochemical stability for long-term operation. More importantly, the nanowire morphology of $N-NiCo_2S_4$ is conserved after the stability test (Fig. 4f, insets). Moreover, XRD and XPS studies also confirm that the structure and surface N of $N-NiCo_2S_4$ could also be maintained after the stability testing (Supplementary Fig. 15 and 16), further indicating its superior structural and chemical stability.

## Discussion

To unravel the effects of N on the electron densities and catalytic properties of $NiCo_2S_4$ at the atomic level, we conducted density functional theory (DFT) calculations. Figure 5a shows the top-view structure of $N-NiCo_2S_4$ (100) with the bond lengths. Detailed structural information of $NiCo_2S_4$ and $N-NiCo_2S_4$ are shown in Supplementary Fig. 17. Overall, the Co–N bond length of 1.86 Å is shorter than the typical Co–S bond (2.23 or 2.20 Å) due to the higher electronegativity of N atoms. In addition, the lengths of Co–S bonds that share the same Co atom with a Co–N bond are also slightly decreased, while all the other nearby Co–S bonds are weakened with the elongation of bond length after N incorporation. These results suggest N doping could effectively modulate the bond strengths in $N-NiCo_2S_4$, which is closely related to the surface electron densities of atoms and the associated absorption/desorption properties. In order to clearly reveal the impacts of the N dopants on the electron densities of the nearby atoms, the electron density differences of $NiCo_2S_4$ and $N-NiCo_2S_4$ are visually mapped in Fig. 5b,c, respectively. Clearly, the electrons of the metal atoms prefer to flow towards nitrogen atoms to form strong Co–N bonds, resulting in nitrogen with more negative charge and a metal with lower electron density. Except for the sulfur atoms of Co–S bonds in which the Co also contains a Co–N bond with slightly higher electron density, the

other nearby sulfur atoms have lower electron densities. These results are consistent with the positive shifts of the binding energies and photo energies of metal and sulfur atoms observed in XPS and XANES spectra. Although the increased electron density around N sites could make the H* unfavorable to desorb and form H*-polluted N sites as sacrificial sites, the decreased electron density of the nearby S atom could facilitate desorption of H* from S sites. Moreover, the d-band centers with a negative shift from −1.93 eV ($NiCo_2S_4$) to −2.05 eV ($N-NiCo_2S_4$) also suggest the weaker chemical adsorption capability with N dopants (Supplementary Fig. 18), which is in good agreement with the experimental UPS results that the valence band shifts far away from the Fermi level after N doping (Supplementary Fig. 8). Although the relationship between the d-band center shift and the reactivity of sulfide surfaces is currently still an open research question, the negatively shifted d-band center basically lowers the energy of the antibonding states, which will weaken the M–S interaction by decreasing the localized electron densities on S sites and the consequent S–H interaction, and thus facilitate the hydrogen desorption from sulfur sites. In addition, the calculated smaller formation energy of substitution (2.08 eV) than that of the insertion state (2.96 eV) further supports the substitutional N is favorable, which is consistent to the XPS and EXAFS results. Furthermore, the calculated density of states (DOS) of bulk $NiCo_2S_4$ and $N-NiCo_2S_4$ (Supplementary Fig. 19) show that the $N-NiCo_2S_4$ has higher electron density at the Fermi level than $NiCo_2S_4$, suggesting that the bulk N most likely plays a role in improving the electrical conductivity of $NiCo_2S_4$. Therefore, the basic working principle of $N-NiCo_2S_4$ is probably to balance the dissociation of $H_2O$ and the desorption of H* by using N dopants to modulate the electronic properties of $NiCo_2S_4$, while the introduced N sites work as sacrificial sites to improve the activity of sulfur sites.

To elucidate the impacts of N dopants on the catalytic process, the energy profiles of the reaction pathway are further calculated

and displayed in Fig. 5d. Typically, hydrogen evolution reaction in alkaline condition involves a two-step pathway. Firstly, the adsorbed $H_2O$ ($H_2O^*$) is cleaved to generate a $H^*$ on the catalyst surface; and then the $H^*$ combines with another H generated by the cleavage of HO–H bond in another $H_2O^*$ molecule to form the adsorbed $H_2$ ($H_2^*$) followed by the desorption of $H_2$ molecule from the catalyst surfaces. Theoretical calculation indicates the $H_2O$ molecule prefers to occupy the top site of Co by the Co–O interaction with an exothermic process, which can be illustrated by electron density difference images in Supplementary Fig. 20. Then, the cleavage of HO–H bond goes through a transition state (TS1) with an energy barrier of 0.74 eV for $NiCo_2S_4$ and 0.56 eV for $N–NiCo_2S_4$ relative to the initial reactant (R1) to form an intermediate (IM1) with $H^*$ on S site and adsorbed OH ($HO^*$) on Co site. Clearly, the incorporation of N could lower the energy barrier of water molecule dissociation reaction on $NiCo_2S_4$. The formed IM1 is followed by desorption of $HO^*$ from the catalyst surface in an exothermic manner, leaving the $H^*$ (IM1') on the sulfur site. The vacated Co top site is then readily available for another $H_2O$ molecule to occupy and form another reactant complex (RC2). In the second step, one of the H atoms in the $H_2O^*$ combines with the $H^*$ on the sulfur site produced in the first step to form an intermediate (IM2), where the generated $H_2^*$ and $HO^*$ occupy on the sulfur site and Co site, respectively. A transition state (TS2) energy barrier, relative to the reactant (R2), must be overcome to achieve this second step, which includes the cleavage of a HO–H bond in $H_2O^*$ and a H–S bond. The energy barrier is 1.63 eV for $NiCo_2S_4$ and 1.61 eV for $N-NiCo_2S_4$, indicating that N dopants cannot only facilitate the water dissociation but also assist to weaken the H–S bond.

Similar to the first step, the $HO^*$ prefers to form $OH^-$ that rapidly migrates from the catalyst surface into the solution, leaving the $H_2^*$ (IM2') on the surface in an exothermic process. Finally, the $H_2^*$ is eliminated to generate free $H_2$ gas and recover its initial state for the next turnover cycle. Overall, N dopants can efficiently facilitate the rate determining steps of water dissociation and H-S breaking, which can be reasonably correlated with the improved HER reaction kinetics in $N–NiCo_2S_4$. Since the reaction energy is also a function of the electrode potential U in the electrocatalytic process, the effect of bias on the two electrochemical elementary steps of HER half reaction ($H_2O + 2e^- \longrightarrow H_2 + 2OH^-$) at pH = 14 is also considered by shifting the chemical potential of the electrons at the equilibrium potential of $U^{SHE} = 0.83$ V, where the reactant and product are at the same energy level. Clearly, the first electrochemical elementary step has become an exothermic reaction at $U^{SHE} = 0.83$ V and the second electrochemical elementary step would proceed on a substantially decreased potential energy surface. Moreover, another surface terminated with Ni atoms is also found to have similar behavior with the surface terminated with $CoS_2$ (Supplementary Fig. 21). Both of surfaces with different terminations all exhibit decreased energy barriers for HER catalysis after nitrogen incorporation.

Given that the desorption of $H^*$ from sulfur sites is a common drawback for conventional metal sulfides for HER reaction, we further implemented this strategy on other commonly used metal sulfides, including $FeS_2$, $CoS_2$, and $NiS_2$ for HER catalysis. Similar to the fabrication process of $N–NiCo_2S_4$, the N doped $FeS_2$ ($N–FeS_2$), $CoS_2$ ($N–CoS_2$), and $NiS_2$ ($N–NiS_2$) were also prepared via the two-step synthesis (see details, Methods). Supplementary Fig. 22 displays the XRD and electrochemical studies of $N–FeS_2$, $N–CoS_2$, and $N–NiS_2$. The XRD patterns of the synthesized metal sulfides and nitrogen-doped counterparts have almost the same profiles, which are in good agreement with the XRD characteristic peaks of $FeS_2$ (JCPDS No. 42–1340), $CoS_2$ (JCPDS No. 65–3322), and $NiS_2$ (JCPDS No. 88-1709), respectively. Interestingly, the electrochemical performance of all the N doped metal sulfides has

been significantly improved (Supplementary Fig. 22). In comparison with the pristine metal sulfides, the overpotentials of $N–FeS_2$, $N–CoS_2$, and $N–NiS_2$ at the current density of 100 mA $cm^{-2}$ are substantially decreased by 160 mV, 120 mV, and 130 mV, respectively. All these results clearly reveal that nitrogen doping could be a facile and general method to promote the HER catalysis of metal sulfides by modulating their surface electronic properties.

In summary, we have demonstrated N doping can significantly improve the HER activity of $NiCo_2S_4$ by rationally manipulating the electron densities of the catalytic sites. XPS, synchrotron-based XANES and DFT calculations consistently elucidate that N dopants can effectively modulate the electronic properties of $NiCo_2S_4$, which could not only help facilitate the $H^*$ desorption from the sulfur sites but also assist the cleavage of HO–H in water molecules. More importantly, the optimized $N–NiCo_2S_4$ NWs achieve an overpotential of 41 mV at 10 mA $cm^{-2}$ and a small Tafel slope of 37 mV $dec^{-1}$, representing the best HER activity so far among the reported metal sulfides, to the best of our knowledge. We also further demonstrated this strategy is a general and facile method to consistently improve the HER catalysis of metal sulfides including $NiCo_2S_4$, $FeS_2$, $CoS_2$, and $NiS_2$. The capability to modulate the electron densities of the catalytic sites could provide valuable insights for the rational design of highly efficient metal sulfides based HER catalysts and beyond.

## Methods

**Materials synthesis**. The NiCo-precursor (Ni–Co–O) NWs were synthesized on Ni foam via a previously developed hydrothermal method[29]. First, the Ni foam substrates were pre-cleaned with diluted HCl solution, deionized (DI) water, and ethanol, respectively. After that, 0.6 mmol $Co(NO_3)_2\cdot6H_2O$, 0.3 mmol Ni $(NO_3)_2\cdot6H_2O$, and 1.5 mmol urea were dissolved in DI water to form a transparent pink colored solution. The prepared solution was then transferred into a 20 mL Teflon-lined stainless-steel autoclave with the pre-cleaned Ni foam ($2 \times 4$ $cm^2$) placed against the wall of Teflon vessel, and the sealed autoclave was further kept at 120 °C in an electric oven for 12 h. After cooling naturally to room temperature, the Ni foam coated with Ni-Co-O NWs was taken out from the autoclaves, alternately washed with DI water and ethanol for several times, and then dried at 60 °C overnight. Finally, $N–NiCo_2S_4$ NWs were fabricated by annealing the as-prepared Ni–Co–O NWs in a home-built tube furnace at 600 °C for 1 h, with S powder as sulfur source placed at the upstream position and ammonia gas as the carrier gas and nitrogen source. Similarly, $NiCo_2S_4$ NWs were synthesized using the Ar gas as the carrier gas and sulfur powder as the sulfur source, and N–NiCoO NWs were obtained under $NH_3$ atmosphere without sulfur source. Similar to the preparation of $N–NiCo_2S_4$, $N–FeS_2$, $N–CoS_2$ and $N–NiS_2$ were also prepared by the two-step method[20,40,41]. Approximately 0.15 M $FeCl_3$ and 1.0 M $NaNO_3$ were mixed in 20 mL a teflon-lined stainless-steel autoclave and kept at 95 °C for 4 h to get FeOOH nanorods. The mixture of 0.1 M $Co(NO_3)_2$, 0.5 M urea, and 0.3 M $NH_4F$ were sealed in the autoclave and maintained at 120 °C for 12 h to get Co(OH)F nanowires. $Ni(OH)_2$ was prepared by the hydrothermal synthetic method with 0.1 M Ni $(NO_3)_2$ and 0.5 M urea as the reaction solution at 120 °C for 12 h. Finally, the prepared precursors were followed by the co-treatment of sulfurization and nitridation at the temperature of 600 °C for 1 h to get the $N–FeS_2$, $N–CoS_2$, and $N–NiS_2$, respectively.

**Materials characterization**. X-ray diffraction (XRD) patterns were recorded on a Philips X'Pert Pro Super diffractometer (with Cu Kα, λ = 1.54182 Å). Field emission scanning electron microscopy (FESEM) conducted on JEOL-JSM-6700F was used to characterize the morphologies of the samples. Transmission electron microscopy (TEM), high resolution transmission electron microscopy (HRTEM), high angle annular dark-field scanning transmission electron microscopy (HAADF-STEM), and energy dispersive X-ray spectroscopy mapping images were collected on Talos F200X and JEMARM 200 F microscope, respectively. The chemical states of nitrogen, cobalt, nickel, and sulfur were measured by X-ray photoelectron spectroscopy (XPS, Thermo Scientific ESCALAB 250Xi) using Mg Kα radiation source of 1253.6 eV, with C 1 s (284.6 eV) as calibration reference. The Co L-edge, Ni L-edge, and S L-edge XANES spectra were collected at the photoemission end-station at beamline BL10B in the National Synchrotron Radiation Laboratory (NSRL) in Hefei, China. The samples were kept in the total electron yield mode under an ultrahigh vacuum at $5 \times 10^{-10}$ mbar. The resolving power of the grating was typically $E/\Delta E = 1000$, and the photon flux was $1 \times 10^{-10}$ photons per second. Spectra were collected at energies from 765 to 805 eV for Co L-edge and 835–880 eV for Ni L-edge in 0.2 eV energy steps. The XANES raw data were normalized by a procedure consisting of several steps. First, the photon energy

was calibrated based on the Au 4 f peak of a freshly sputtered gold wafer and then substrate a line to set the pre-edge at zero. Finally, the spectra were normalized to yield an edge-jump to one. In the ultraviolet photoemission spectroscopy (UPS) measurements, an excitation energy of 169.08 eV was utilized. The extend X-ray absorption fine structure (EXAFS) spectra were collected at 14W1 station in Shanghai Synchrotron Radiation Facility (SSRF) and 1W1B station in Beijing Synchrotron Radiation Facility (BSRF). EXAFS measurements at the Co K-edge was performed in a transmission mode, using ionization chambers with optimized detecting gases to measure the radiation intensity.

**Electrochemical characterization**. The electrochemical measurements were carried out on the CHI760E electrochemical workstation using a three-electrode system in 1.0 M KOH aqueous solution. The studied materials grown on Ni foams served directly as the working electrode, while graphite rod and Ag/AgCl (saturated KCl solution) were used as the counter electrode and the reference electrode, respectively. The control sample of Pt/C was prepared by drop-coating Pt/C catalyst ink on Nickel foam (2.3 mg cm$^{-2}$). The Pt/C catalyst ink was prepared by homogeneously dispersing 20 mg Pt/C (20 wt%) and 10 μL of 5 wt% Nafion solution in 1 mL water/isopropanol (4:1 v/v) solution. All potentials measured versus Ag/AgCl were converted to reversible hydrogen electrode (RHE), using the following equation: $E$(vs. RHE) = $E$ (vs. Ag/AgCl) + 1.023 V in the 1.0 M KOH. Polarization curves and Tafel curves were recorded at the scan rate of 5 mV s$^{-1}$. Electrochemical impedance spectroscopy (EIS) measurements were conducted at the overpotential of 50 mV with a potential perturbation of 5 mV amplitude in the range from 10 kHz to 0.1 Hz. The electrolyte resistance ($R_s$) was measured using EIS measurements and used for iR compensation by the equation of $E_{iR-corrected}$ = $E_{original}$−$I × R_s$[42]. The electrochemical surface areas (ECSAs) were estimated by cyclic voltammograms (CVs) at the potential range from 0.1 V to 0.2 V at the scan rates of 20, 40, 60, and 80 mV s$^{-1}$, respectively[43]. The current density differences ($Δj = j_a$−$j_c$) were plotted against scan rates, the linear slope of which, twice the values of $C_{dl}$ used to estimate the ECSA.

**Density functional theory calculation**. Density functional theory (DFT) calculation was performed using CASTEP code as implemented in the Materials Studios package of Accelrys Inc[44]. The electron exchange-correlation potential was conducted by the Perdew-Burke-Ernzerhof (PBE) functional of generalized gradient approximation (GGA). The ultrasoft pseudopotentials was employed and the core electrons of atoms were treated using effective core potential (ECP). The kinetic energy cutoff was set to 400 eV for the plane-wave basis set and the DFT dispersion correction (DFT-D) method was used to treat the van der Waals interactions. Brillouin zone integration was sampled with the 5 × 5 × 5 and 2 × 1 × 1 Monkhorst-Pack mesh k-point for bulk and surface calculations, respectively. The optimized lattice parameters of the bulk crystal were $a = b = c = 9.379$ Å for NiCo$_2$S$_4$, in good agreement with experimental values of $a = b = c = 9.387$ Å. The bulk N-doped-NiCo$_2$S$_4$ crystal was constructed by replacing S atom with N atom with a doping percentage of about 3%. The CoS$_2$-terminated NiCo$_2$S$_4$ (100) or Ni-terminated NiCo$_2$S$_4$ (100), the most common surface for spinel materials, as a predominant growth surface was modeled by a periodic eight-layer slab repeated in √2×√2 surface unit cell with a vacuum region of 10 Å between the slabs along the Z axis[45–47]. The convergence tolerances were set to 5.0 × 10$^{-6}$ eV per atom for energy, 5.0 × 10$^{-4}$ Å for maximum displacement, and 0.01 eV Å$^{-1}$ for maximum force. The adsorbates together with the top six layers of surfaces atoms were relaxed, while the two bottom layers were constrained at the bulk position during the geometry optimizations. A complete LST/QST approach was used to determine the transition state. The formation energy ($E_{form}$) of the substitution or insertion on the base of the following formula: $E_{form} = E_{tot}$ (substituted)−$E_{tot}$ (pristine)−$μ_N$ + $μ_S$; $E_{form} = E_{tot}$ (inserted)−$E_{tot}$ (pristine)−$μ_N$, where the $E_{tot}$ (substituted), $E_{tot}$ (inserted), and $E_{tot}$ (pristine) are the total energies of N-substituted, N-inserted, and pristine NiCo$_2$S$_4$, respectively. The $μ_N$ and $μ_S$ denote the chemical potentials of nitrogen and sulfur atoms, respectively. The $μ_N$ is the half of the total energy of an isolated ground-state N$_2$ molecule, while the $μ_S$ is obtained from the S$_8$ molecule. The Volmer-Heyrovsky pathway of HER in alkaline 2H$_2$O + 2e$^-$ ⟶ H$_2$ + 2OH$^-$ proceeds through two electrochemical elementary steps with two electrons involved including H$_2$O + * + e$^-$ ⟶ H* + OH$^-$ (1) and H* + H$_2$O + e$^-$ ⟶ H$_2$ + * + OH$^-$ (2), where the * denotes the catalyst surface. The chemical potential for the reaction (H$^+$ + e$^-$) could be related to that of 1/2H$_2$−$k_B T$ln10 × pH, when setting the reference potential to be that of the standard hydrogen electrode (SHE) with different pH. The chemical potential ΔG(OH$^-$) could be derived from the G(H$_2$O)−G(H$^+$), that is ΔG(OH$^-$) = G(H$_2$O)−G(H$^+$) = G(H$_2$O)−G(1/2H$_2$) + $k_B T$ln10 × pH ($k_B$ is Boltzmann's constant) at an electrode potential U = 0 vs. SHE. Therefore, the corresponding free energies of the above two electrochemical elementary steps could be calculated as follows: ΔG$_1$ = G(H*) + G(1/2H$_2$)−G(*) + $k_B T$ln10 × pH and ΔG$_2$ = G(*) + G(1/2H$_2$)−G(H*) + $k_B T$ln10 × pH, respectively, which correspond with the energy differences of IM1' and R1, and P and R2, respectively. The free energies of all intermediates at the applied electrode equilibrium potential U could be calculated by G(U) = G—neU, where n is the electron number of each step[48].

**Data availability**. The authors declare that the main data supporting the findings of this study are available within the article and its Supplementary Information. Extra data are available from the corresponding author upon request.

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

# ARTICLE

26. Wang, H. T. et al. Transition-metal doped edge sites in vertically aligned MoS₂ catalysts for enhanced hydrogen evolution. *Nano Res.* **8**, 566–575 (2015).

27. Wu, J. H. et al. One-step hydrothermal synthesis of NiCo₂S₄-rGO as an efficient electrocatalyst for the oxygen reduction reaction. *J. Mater. Chem. A* **2**, 20990–20995 (2014).

28. Liu, Q., Jin, J. T. & Zhang, J. Y. NiCo₂S₄@graphene as a bifunctional electrocatalyst for oxygen reduction and evolution reactions. *ACS Appl. Mater. Interfaces* **5**, 5002–5008 (2013).

29. Sivanantham, A., Ganesan, P. & Shanmugam, S. Hierarchical NiCo₂S₄ nanowire arrays supported on Ni foam: an efficient and durable bifunctional electrocatalyst for oxygen and hydrogen evolution reactions. *Adv. Funct. Mater.* **26**, 4661–4672 (2016).

30. An, Y. et al. Constructing three-dimensional porous Ni/Ni₃S₂ nano-interfaces for hydrogen evolution electrocatalysis under alkaline conditions. *Dalton. Trans.* **46**, 10700–10706 (2017).

31. Sun, Z. H. et al. Graphene activating room-temperature ferromagnetic exchange in cobalt-doped ZnO dilute magnetic semiconductor quantum dots. *Acs Nano* **8**, 10589–10596 (2014).

32. Lu, X. H. et al. High energy density asymmetric quasi-solid-state supercapacitor based on porous vanadium nitride nanowire anode. *Nano. Lett.* **13**, 2628–2633 (2013).

33. Shen, L. F. et al. NiCo₂S₄ nanosheets grown on nitrogen-doped carbon foams as an advanced electrode for supercapacitors. *Adv. Energy Mater.* **5**, 1400977 (2015).

34. Zhai, T. et al. An electrochemical capacitor with applicable energy density of 7.4 Wh/kg at average power density of 3000 W/kg. *Nano. Lett.* **15**, 3189–3194 (2015).

35. Fleet, M. E. Xanes spectroscopy of sulfur in earth materials. *Can. Mineral.* **43**, 1811–1838 (2005).

36. Jasion, D. et al. Low-dimensional hyperthin FeS₂ nanostructures for efficient and stable hydrogen evolution electrocatalysis. *Acs Catal.* **5**, 6653–6657 (2015).

37. Caban-Acevedo, M. et al. Efficient hydrogen evolution catalysis using ternary pyrite-type cobalt phosphosulphide. *Nat. Mater.* **14**, 1245–1251 (2015).

38. Xu, K. et al. Regulating water-reduction kinetics in cobalt phosphide for enhancing HER catalytic activity in alkaline solution. *Adv. Mater.* **29**, 1606980 (2017).

39. Staszak-Jirkovsky, J. et al. Design of active and stable Co-Mo-Sₓ chalcogels as pH-universal catalyst for the hydrogen evolution reaction. *Nat. Mater.* **15**, 197–203 (2016).

40. Ling, Y. C., Wang, G. M., Wheeler, D. A., Zhang, J. Z. & Li, Y. Sn-doped hematite nanostructures for photoelectrochemical water splitting. *Nano Lett.* **11**, 2119–2125 (2011).

41. Chen, P. Z. et al. Metallic Co₄N porous nanowire arrays activated by surface oxidation as electrocatalysts for the oxygen evolution reaction. *Angew. Chem. Int. Ed.* **54**, 14710–14714 (2015).

42. Chen, G. F. et al. Efficient and stable bifunctional electrocatalysts Ni/NiₓMᵧ (M=P, S) for overall water splitting. *Adv. Funct. Mater.* **26**, 3314–3323 (2016).

43. Tang, C. et al. Ternary FeₓCo₁₋ₓP nanowire array as a robust hydrogen evolution reaction electrocatalyst with Pt-like activity: experimental and theoretical insight. *Nano Lett.* **16**, 6617–6621 (2016).

44. Clark, S. J. et al. First principles methods using CASTEP. *Z. Krist.* **220**, 567–570 (2005).

45. Bliem, R. et al. Subsurface cation vacancy stabilization of the magnetite (001) surface. *Science* **346**, 1215–1218 (2014).

46. Bliem, R. et al. Cluster nucleation and growth from a highly supersaturated adatom phase: silver on magnetite. *ACS Nano* **8**, 7531–7537 (2014).

47. Pham, H., Cheng, M., Frei, H. & Wang, L. Surface proton hopping and fast-kinetics pathway of water oxidation on Co₃O₄ (001) Surface. *ACS Catal.* **6**, 5610–5617 (2016).

48. Zheng, Y. et al. Nanoporous graphitic-C₃N₄@carbon metal-free electrocatalysts for highly efficient oxygen reduction. *J. Am. Chem. Soc.* **133**, 20116–20119 (2011).

## Acknowledgements

We thank the financial support of the Natural Science Fund of China (Nos. 21771169 and 11722543), USTC start-up funding, Recruitment Program of Global Expert, and the Fundamental Research Funds for the Central Universities (WK2060190074; WK2060190081). We also appreciate the Photoemission Endstations (BL10B) in National Synchrotron Radiation Laboratory (NSRL) for help in XANES and XAFS characterizations. The numerical calculations in this paper have been done on the supercomputing system in the Supercomputing Center of University of Science and Technology of China.

## Author contributions

G.M.W. designed and supervised the project. Y.S.W. and X.J.L. conducted the project and contributed equally to this work. X.S.Z. performed the XANES, EXAFS, and UPS measurements. X.H.X. and X.Y.S. provided the XPS measurements. Y.S.W., D.D.H., Y. S., S.W.N., and Z.Y.C. helped to analyze the data. L.S. proformed HRTEM and element mapping. Y.S.W., Y.F.X., J.Y.C., S.Y.W., and J.K. helped in synthesizing samples. J.B.Z. offered SEM test. G.M.W., Y.S.W., and X.J.L. wrote and revised the manuscript.

## Additional information

**Competing interests:** The authors declare no competing interests.

