## [Peer Review File · Nature Communications]

Reviewers' comments:

Reviewer #1 (Remarks to the Author):

Recommendation:

Modified Sulfides and other alternative materials for hydrogen evolution is a very active field and substituting Pt is of interest.

I recommend major revisions, because the present paper tries to do 3 different things and does none of them at the standards required for Nature communications. They find a new highly active N-doped material, but fail to prove that it is stable. Secondly, they show that one can synthesize other N doped sulfides and that this lower the HER overpotential but provide no evidence of stability, and thirdly they aim to establish why N doped materials have improved activity. Their structure and activity correlation is found by the use of XPS and XANES to show that the electronic structure is modified and then DFT to show that this matters for the surface reactivity. However, the DFT study does not provide any evidence because either it uses the wrong references or the wrong surface terminations or a combination of both. However, I feel that with the methods that the authors have used they can provide stronger evidence for why N doping provides improved HER activity which is why I recommend major revisions.

The authors have 3 main claims in the abstract

Claim 1:

“Demonstrate a novel and general strategy to effectively modulate the electron densities around the sulfur sites in NiCo₂S₄ and other commonly used metal sulfides for HER catalysis. “

This claim is supported by the provided XPS and Xanes results

Claim 2:

“Instead of directly weakening the H-S interaction sacrifice part of the sulfur sites by N-substitution”

I do not find that the authors provide evidence that N substitutes at S sites. They provide evidence that N is part of the structure but no evidence is provided that N is incorporated by substitutional doping of S sites. There are several ways to provide more support for this claim, i.e. a more detailed DFT study of the possible sites, a more careful XRD analysis might show changes in lattice parameters, adding EXAFS analysis to the XANES results, and maybe if available to the authors a higher resolution TEM study of the S and N distribution.

Claim 3: “Which could indirectly decrease the electron densities around the remaining sulfur sites and thus help facilitate the H* desorption.”

This claim is backed by XPS, and XANES, indicating that also bulk S is modified by N doping. The XANES results gives the average of the bulk and the surface of the nanowires, while XPS gives the surface composition, could the authors please comment on what the role of the bulk N is and evidently it would be interesting with respect to stability and the claimed effect of N to determine if the surface N is stable?

However, the authors cause and effect explanation is a little puzzling. The authors find that the d-band centers are modulated by N doping to a lower d-band center and that the metal S-bond is weakened. Why does this lead to the S being less reactive? If S bind more weakly to the metal atoms it would reversely bond stronger to the H atoms [see e.g. Eva M Fernández et al, *Angewandte Chemie International Edition* 2008, 47, 4683], giving the opposite effect than the explanation provided by the authors.

Major concerns

The DFT study of the paper is flawed.

For instance the authors write: “Then, the cleavage of HO-H bond goes through a transition state of TS1 with an energy barrier of 1.32 eV for NiCo₂S₄ and 1.06 eV for N-NiCo₂S₄ to form an intermediate product of IM1 with H* on S site and adsorbed OH (HO*) on Co site.”

The relevance of the calculation of barriers are unclear since it is not an electrochemical elementary step (no electron or charge species are involved). The non-electrochemical elementary step will not play a role since the barrier of the reported reaction is much too high to give any turnover at the experimental conditions applied at HER. The authors have to account for the potential of the electron and describe how they determine the energy of the OH⁻ formed. Based on figure 5 one would conclude that this surface cannot evolve H₂, just the thermodynamical energy difference between RC1 and IM1 and RC2 and IM2 is so high that these intermediate structure are irrelevant. The final structures with the H* and OH⁻ are on the other hand much more negative than the initial structure that one suspects that the authors are not studying the relevant stable surfaces under reaction conditions.

To make the paper suitable to publication the authors have to extend the DFT study so that it can model the relevant reactions. Furthermore, I would suggest to also use DFT to support the claim that N substitutes S at S sites.

Stability:

On page 12 line 232-237 the “long term stability” is tested showing that over 40000seconds 90% of initial current density is retained. Long term stability for electrolysis systems is measured in units of 1000 of hours. Thus, the provided data is not a proof of long term stability, for instance does this loss continue? In addition, a more detailed structural characterization than the inserted SEM images at micrometer resolution is needed to provide evidence of structural stability.

Reviewer #2 (Remarks to the Author):

Wu et al. have demonstrated N incorporation could effectively modulate the electron densities of transition metal sulfides and substantially improve the HER catalysis of NiCo₂S₄ and other metal sulfides. Detailed structural, electrochemical and theoretical characterization have been conducted to clearly elucidate the effects of N on the electronic and catalytic properties of NiCo₂S₄. Overall, this work is interesting and significant, and could provide valuable insights for the design of HER catalysts in future. Thus, I would like to recommend it for publication on Nature Communications after minor revision.

1. The NAXES spectrum is a useful technique to characterize the electron density variation. Could the authors give more discussion on it?
2. Could the author give more details on the NAXES measurement and the normalization details of the Co and Ni L-edge.
3. Please label the diffraction peaks in the XRD patterns in Figure 2 and Figure S12.
4. The authors obtained a pretty good catalytic performance on N-NiCo₂S₄ with an overpotential of 41 mV at 10 mA cm⁻². Please compare the HER performance with other metal sulfides reported in literature and make a comparison table in the supplementary information.

Reviewer #3 (Remarks to the Author):

The manuscript introduces the introduction of NH₃ to the sulfurization of NiCo₂S₄ and the catalytic activity of product (N-NiCo₂S₄) in hydrogen evolution reaction (HER). The performance of N-NiCo₂S₄ is better than that of NiCo₂S₄, and close to that of commercial Pt/C electrocatalyst. The synthesis is convenient, the performance is promising, and the electrochemical characterization is clear. The result would contribute positively to the development of low-cost clean and renewable energy. However, I cannot suggest the acceptance of manuscript because of the following reasons.

1. The structure characterization of sample needs to be improved. For example, the authors claimed the N was homogeneous in N-NiCo₂S₄, while EDS mapping shows that the signal distributions of Ni, Co, and N cannot overlap each other, implying that the distribution of Ni, Co, and N is inhomogeneous. On the other hand, XRD is hard to reveal small amount of other possible structure in the product. Therefore, EDS mapping with much large magnification (for example, atomic resolution EDS mapping) is necessary to reveal what is the real microstructure of the sample.

2. The authors claimed that the introduction of N improved the intrinsic catalytic activity of N-NiCo₂S₄. The TOFs of N-NiCo₂S₄ and NiCo₂S₄ should be determined to support this claim.
3. The XPS spectra should be interpreted in more details. The deconvolution of XPS peaks should be carried out to find out all possible chemical state of Ni, Co, S, N, and O. The sulfide is easy to be oxidized due to air exposure. Actually, strong oxygen peak can be found from the survey XPS spectrum (Figure S4), suggesting the presence of nickel oxide or/and cobalt oxide. The contribution of these oxides to the HER process should be evaluated.
4. The atomic ratio of Ni, Co, S, N and O should be offered. The possibility of defect formation and the consequent influence on the performance should be discussed.
5. The loading amount of N-NiCo₂S₄ should be introduced. Meanwhile, Pt/C should be loaded onto Ni foam with the same loading amount as N-NiCo₂S₄ for the convincing performance comparison. On the other hand, the table that lists the performance of reported electrocatalysts is required.
6. The authors claimed that “The CoS₂-terminated NiCo₂S₄ (1 0 0) as a predominant growth surface was modeled...”. However, not experimental data showed the predominant growth surface of NiCo₂S₄ is (100) plane.
7. The computation result showed that the d band center of N-NiCo₂S₄ is shifted far from the Fermi level. This should be confirmed by valence band XPS of UPS experiment.

Point-by-point response to the referees' comments

We sincerely thank the referees for their careful review and valuable comments, which certainly help improve our manuscript. We also appreciate the opportunity the editor gives us, to address the comments and revise the manuscript. Our point-by-point responses are presented below and all the changes in the revised manuscript have been highlighted in yellow for your review.

Reviewer #1 (Remarks to the Author):

Recommendation:

Modified Sulfides and other alternative materials for hydrogen evolution is a very active field and substituting Pt is of interest. I recommend major revisions, because the present paper tries to do 3 different things and does none of them at the standards required for Nature communications. They find a new highly active N-doped material, but fail to prove that it is stable. Secondly, they show that one can synthesize other N doped sulfides and that this lower the HER overpotential but provide no evidence of stability, and thirdly they aim to establish why N doped materials have improved activity. Their structure and activity correlation is found by the use of XPS and XANES to show that the electronic structure is modified and then DFT to show that this matters for the surface reactivity. However, the DFT study does not provide any evidence because either it uses the wrong references or the wrong surface terminations or a combination of both. However, I feel that with the methods that the authors have used they can provide stronger evidence for why N doping provides improved HER activity which is why I recommend major revisions.

Response: We sincerely thank the reviewers for the opportunity you kindly give us to revise the manuscript, and we also appreciate your valuable comments to improve our manuscript. Following your suggestions and comments on the stability and the DFT calculations, we have conducted more experiments and made point-by-point response to your comments, which is presented below.

The authors have 3 main claims in the abstract

Claim 1:

“Demonstrate a novel and general strategy to effectively modulate the electron densities around the sulfur sites in NiCo₂S₄ and other commonly used metal sulfides for HER catalysis. “This claim is supported by the provided XPS and Xanes results.

Claim 2:

“Instead of directly weakening the H-S interaction sacrifice part of the sulfur sites by N-substitution”

I do not find that the authors provide evidence that N substitutes at S sites. They provide evidence that N is part of the structure but no evidence is provided that N is incorporated by substitutional doping of S sites. There are several ways to provide more support for this claim, i.e. a more detailed DFT study of the possible sites, a more careful XRD analysis might show changes in lattice parameters, adding EXAFS

analysis to the XANES results, and maybe if available to the authors a higher resolution TEM study of the S and N distribution.

Response: We warmly thank the reviewer for the valuable comments. According to the reviewer's advices, we performed both experiments and DFT calculations to support the claim that N substitutes S at the S sites. Firstly, the feasibility of N incorporated in NiCo₂S₄ was been investigated by calculating the formation energy (E_{form}) of the substitution or insertion on the base of the following formula:

$$E_{\text{form}}=E_{\text{tot}}(\text{substituted})-E_{\text{tot}}(\text{pristine})-\mu_{\text{N}}+\mu_{\text{S}};$$

$$E_{\text{form}}=E_{\text{tot}}(\text{inserted})-E_{\text{tot}}(\text{pristine})-\mu_{\text{N}};$$

where the $E_{\text{tot}}(\text{substituted})$, $E_{\text{tot}}(\text{inserted})$, and $E_{\text{tot}}(\text{pristine})$ are the total energies of the N-substituted, N-inserted, and pristine NiCo₂S₄, respectively. The μ_{N} and μ_{S} denote the chemical potentials of nitrogen and sulfur atoms, respectively. The μ_{N} is the half of the total energy of an isolated ground-state N₂ molecule, while the μ_{S} is obtained from the S₈ molecule. The calculated formation energies for the N substitution and insertion states are 2.08 eV and 2.91eV, respectively, suggesting that it would be more favorable for N to substitute the S sites. Additionally, we also performed careful XRD and EXAFS analysis. Figure R1 shows the magnified (4 0 0) diffraction peak with a slight shift towards the higher diffraction angle, which means the crystal lattice of NiCo₂S₄ is compressed after the incorporation of nitrogen. Meanwhile, the DFT calculated lattice parameters for NiCo₂S₄, N-substituted-NiCo₂S₄, and N-inserted-NiCo₂S₄ are 9.372 Å, 9.329 Å, and 9.407 Å, respectively. Clearly, the N substitution state in NiCo₂S₄ is consistent to the XRD results with the decreased crystal lattice parameters. Furthermore, EXAFS measurements were performed to study the doping state of N. As shown in Figure R2a, the slight shift of Co k-edge $k^3\chi(k)$ oscillation to lower k-region, implies that the coordination number of Co-S reduces because N atoms occupy S sites to form Co-N. Meanwhile, the corresponding Fourier transform (FT) curves are shown in Figure R2b. The decreased intensity of Co-S peak in the Co R space suggests that the coordination of S-Co reduces, probably due to the replacement of S by N. In addition, XPS N 1s in Figure S6c also reveal the existence of N-metal bond, further suggesting the N substitution. For high resolution TEM, it is difficult to directly observe the atomic difference of N and S in atomic resolution, because it is beyond the limit of our TEM instrument. Therefore, we conducted TEM-element mapping with higher magnification which also displays uniform element distribution of N and S in N-NiCo₂S₄ (Figure R3). Taken together, these results clearly support that N is incorporated by replacing the S sites. We have added the related data in the revised manuscript and give corresponding discussion on it.

Figure R1 | The XRD patterns of NiCo_2S_4 and $\text{N-NiCo}_2\text{S}_4$. Inset: the magnified curves showing a slight shift in the (4 0 0) peak position towards the higher angle region.

Figure R2 | (a) k^3 -weighted $\chi(k)$ spectra and (b) Fourier transformed Co K-edge EXAFS spectra for NiCo_2S_4 (blue) and $\text{N-NiCo}_2\text{S}_4$ (red).

Figure R3 | The element mapping images of Ni, Co, S and N elements of $\text{N-NiCo}_2\text{S}_4$ NW with a higher magnification in Figure 2g; the scale bar is 10 nm.

Claim 3: “Which could indirectly decrease the electron densities around the remaining sulfur sites and thus help facilitate the H^* desorption.”

This claim is backed by XPS, and XANES, indicating that also bulk S is modified by N doping. The XANES results gives the average of the bulk and the surface of the nanowires, while XPS gives the surface composition, could the authors please comment on what the role of the bulk N is and evidently it would be interesting with respect to stability and the claimed effect of N to determine if the surface N is stable?

Response: We thank the reviewer for the insightful comments. We agree that XPS gives the surface information such as compositions and chemical states, and the synchrotron based XANES can get the average information. Actually, Both XPS and XANES results in Figure 3 in the manuscript show consistent trends in the chemical information evolution after nitrogen incorporation, indicating N doping can change both surface and bulk electronic properties of NiCo_2S_4 . The bulk N could mainly contribute to improve the electrical conductivity of the NiCo_2S_4 , which has been supported by the DFT calculation (Figure R4). The calculated density of state (DOS) of bulk NiCo_2S_4 and N- NiCo_2S_4 show that the N- NiCo_2S_4 has slightly higher electron densities at Fermi level than NiCo_2S_4 , suggesting that the bulk N probably plays the role in improving the conductivity of NiCo_2S_4 . To further verify the stability of the surface N, we performed XPS studies on the N- NiCo_2S_4 after the long-term catalytic test. After testing, we still observed the existence of N on the surface of NiCo_2S_4 , indicating the surface N is very stable during electrocatalytic hydrogen evolution (Figure R5).

Figure R4 | Density of state (DOS) plots of d orbital contribution to the bulk NiCo_2S_4 and bulk N- NiCo_2S_4 . The dashed line is Fermi level.

Figure R5 | XPS N 1s of N-NiCo₂S₄ before and after stability test.

However, the authors cause and effect explanation is a little puzzling. The authors find that the d-band centers are modulated by N doping to a lower d-band center and that the metal S-bond is weakened. Why does this lead to the S being less reactive? If S bind more weakly to the metal atoms it would reversely bond stronger to the H atoms [see e.g. Eva M Fernández et al, *Angewandte Chemie International Edition* 2008, 47, 4683], giving the opposite effect than the explanation provided by the authors.

Response: We appreciate the reviewer’s comments and we are pleased to clarify this issue. The d-band centers are modulated by N doping to a lower d-band center, and thus the metal-S bond is weakened. The weakened metal-S bond results in the decreased localized electron cloud densities on S site which is supported by our XPS and XANES studies. Thus, it weakens the S-H interaction and facilitates desorption of hydrogen to promote the HER activity, due to the decreased electron density on sulfur sites. For the reference of *Angew. Chem. Int. Ed.*, **47**, 4683 (2008), they presented the adsorptions of S and SH on different metal sulfide surfaces and showed a scaling relationship between the adsorption energies of S and SH on the metal sulfide surface, where the stronger interaction between S adsorbate and metal sulfide surface correlates the stronger binding between SH adsorbate and the metal sulfide surface, but they didn’t relate the interaction between M-S and S-H.

Major concerns

The DFT study of the paper is flawed.

For instance the authors write: “Then, the cleavage of HO-H bond goes through a transition state of TS1 with an energy barrier of 1.32 eV for NiCo₂S₄ and 1.06 eV for N-NiCo₂S₄ to form an intermediate product of IM1 with H* on S site and adsorbed OH (HO*) on Co site.”

The relevance of the calculation of barriers are unclear since it is not an electrochemical elementary step (no electron or charge species are involved). The non-electrochemical elementary step will not play a role since the barrier of the reported reaction is much too high to give any turnover at the experimental conditions

applied at HER. The authors have to account for the potential of the electron and describe how they determine the energy of the OH⁻ formed. Based on figure 5 one would conclude that this surface cannot evolve H₂, just the thermo dynamical energy difference between RC1 and IM1 and RC2 and IM2 is so high that these intermediate structure are irrelevant. The final structures with the H* and OH⁻ are on the other hand much more negative than the initial structure that one suspects that the authors are not studying the relevant stable surfaces under reaction conditions. To make the paper suitable to publication the authors have to extend the DFT study so that it can model the relevant reactions. Furthermore, I would suggest to also use DFT to support the claim that N substitutes S at S sites.

Response: We truly appreciate the reviewer's comments and suggestions, which greatly improve the understanding of the HER catalytic process. Following the suggestions, we have performed the extended calculation and also investigated the surface terminated with Ni.

First, the Volmer-Heyrovsky pathway of hydrogen evolution reaction in alkaline $2\text{H}_2\text{O} + 2\text{e}^- \rightarrow \text{H}_2 + 2\text{OH}^-$ proceeds through two electrochemical elementary steps and two electrons involved.

where the * denotes the catalyst surface. The chemical potential for the reaction ($\text{H}^+ + \text{e}^-$) could be related to that of $1/2\text{H}_2 - k_{\text{B}}T\ln 10^{\text{pH}}$, when setting the reference potential to be that of the standard hydrogen electrode (SHE) with different pH. The chemical potential $\Delta G(\text{OH}^-)$ could be derived from the $G(\text{H}_2\text{O}) - G(\text{H}^+)$, that is $\Delta G(\text{OH}^-) = G(\text{H}_2\text{O}) - G(\text{H}^+) = G(\text{H}_2\text{O}) - G(1/2\text{H}_2) + k_{\text{B}}T\ln 10^{\text{pH}}$ (k_{B} is Boltzmann's constant) at an electrode potential $U = 0$ vs. SHE. Therefore, the corresponding free energies of the above two electrochemical elementary steps could be calculated as followed:

$$\Delta G_1 = G(\text{H}^*) - G(1/2\text{H}_2) - G(*) + k_{\text{B}}T\ln 10^{\text{pH}};$$

$$\Delta G_2 = G(*) + G(1/2\text{H}_2) - G(\text{H}^*) + k_{\text{B}}T\ln 10^{\text{pH}};$$

Figure R6 shows the relative energy profiles and the simplified surface structures of the various reaction species along the reaction pathway. Clearly, the difference energy ΔG_1 and ΔG_2 corresponded with the thermodynamic energy level of IM1' (0.64/0.53 eV of NiCo₂S₄/N-NiCo₂S₄) relative to that of R1(2) and P (1.66 eV) relative to that of R2 (0.64/0.53 eV of NiCo₂S₄/N-NiCo₂S₄), respectively. The whole reaction pathway is an uphill state and the reaction is bound to overcome high energy barrier to proceed. However, in the electrocatalytic processes, due to the inclusion of electrons on the electrode, the energy levels of adsorbed intermediates are also the functions of the electrode potential U . The free energies of all intermediates at the applied electrode equilibrium potential U could be calculated by $G(U) = G - neU$, where n is the transfer electron number of each step. The equilibrium potential U^0 for HER half reaction $2\text{H}_2\text{O} + 2\text{e}^- \rightarrow \text{H}_2 + 2\text{OH}^-$ at pH=14 was determined to be 0.83 V vs SHE, where the reactant and product are at the same energy level. Therefore, we consider the effect of this bias on the two electrochemical elementary steps by shifting the

chemical potential of the electrons at the equilibrium potential of $U^{\text{SHE}} = 0.83\text{V}$. As shown in Figure 5, the three thermodynamic energy levels have been shifted to be 0, -0.19/-0.30 (NiCo₂S₄/N-NiCo₂S₄) at equilibrium potential $U^{\text{SHE}} = 0.83\text{V}$, respectively, and the corresponding thermodynamic energy barriers are also shifted down with 0.83 eV, which thus leads to the substantially decreased energy barriers.

In addition, the surface terminated with Ni atoms was also investigated. There are two different water adsorption sites including Ni and Co on this surface, while the Ni sites has higher energy barrier for water dissociation (1.43 eV) than the Co sites (1.20 eV). Figure R7 presented the favorable reaction pathway of water dissociation on the Co sites of Ni terminated surface. The energy barriers of the water dissociation on Co site are 0.56 eV (N-NiCo₂S₄) and 0.69 eV (NiCo₂S₄) relative to the initial reactant (R1) at potential $U = 0\text{V}$, which are very close to those of pathway proceeded on the CoS₂ terminated surface with 0.56 eV (N-NiCo₂S₄) and 0.74 eV (NiCo₂S₄), respectively. More importantly, we can see that both surfaces with different surface termination all exhibit decreased energy barriers for HER catalysis after nitrogen incorporation. We have added the extended results in the revised manuscript and given corresponding discussion.

Figure R6 | DFT calculations of NiCo₂S₄ and N-NiCo₂S₄. (a) The top-view structures of N-NiCo₂S₄ (1 0 0) with labeled bond lengths of Co-S and Co-N bonds. The top-view electron density difference of (b) NiCo₂S₄ (1 0 0) and (c) N-NiCo₂S₄ (1 0 0). (d) Relative energy profiles and the simplified surface structures of the various reaction species along the reaction pathway, including the H* formation process (left panel) and H₂ formation process (right panel) in alkaline media on the NiCo₂S₄ (1 0 0) and N-NiCo₂S₄ (1 0 0) at electrode potential $U = 0\text{V}$ and $U^{\text{SHE}} = 0.83\text{V}$, respectively.

Figure R7 | Relative energy profiles of the H* formation process on Ni terminated surface in alkaline media on the NiCo₂S₄ (1 0 0) and N-NiCo₂S₄ (1 0 0) at electrode potential U = 0 V and U^{SHE} = 0.83V, respectively.

Stability:

On page 12 line 232-237 the “long term stability” is tested showing that over 40000seconds 90% of initial current density is retained. Long term stability for electrolysis systems is measured in units of 1000 of hours. Thus, the provided data is not a proof of long term stability, for instance does this loss continue? In addition, a more detailed structural characterization than the inserted SEM images at micrometer resolution is needed to provide evidence of structural stability.

Response: Following the suggestions, we have conducted a long-term test to evaluate the stability of N-NiCo₂S₄. After a continuous 1000 h test, ~ 68.2% of the initial current density can be maintained (Figure R8), which is still much better than most of the reported HER catalysts in literature with only less than 100 hours testing. In addition, we also observed a small amount of catalysts on nickel foam fall off after the long-term HER test, which may be part of the reason for the catalytic loss. In order to verify the structural and morphological stability of N-NiCo₂S₄, we further conducted SEM, XRD and XPS measurements after long-term testing (Figure R8, R9 and R5). SEM image indicates the nanowire morphology can be kept after the long-term stability test. In addition, XRD and XPS analysis further confirm that the structure and surface N of N-NiCo₂S₄ can be well maintained after the stability testing. (Figure R9 and R5). Taken together, all these results clearly prove the prepared N-NiCo₂S₄ own excellent catalytic stability for HER catalysis. We have added these data in the revised manuscript and given corresponding discussion on it.

Figure R8 | Current density versus time ($i-t$) curve of N-NiCo₂S₄ NWs recorded for 1000 h at 50 mV vs. RHE, without IR correlation. The SEM images for N-NiCo₂S₄ before and after the stability test are illustrated as the insets.

Figure R9 | The XRD patterns of N-NiCo₂S₄ before and after stability test.

Reviewer #2 (Remarks to the Author):

Wu et al. have demonstrated N incorporation could effectively modulate the electron densities of transition metal sulfides and substantially improve the HER catalysis of NiCo₂S₄ and other metal sulfides. Detailed structural, electrochemical and theoretical characterization have been conducted to clearly elucidate the effects of N on the electronic and catalytic properties of NiCo₂S₄. Overall, this work is interesting and significant, and could provide valuable insights for the design of HER catalysts in future. Thus, I would like to recommend it for publication on Nature Communications after minor revision.

Response: We appreciate the reviewer's positive comments and recommendation for publication in *Nature Communications*.

1. The XANES spectrum is a useful technique to characterize the electron density variation. Could the authors give more discussion on it?

Response: XANES are commonly used to probe the electronic information of the materials. By synchronizing the irradiation of the electron, the electronic transition information can be obtained by the change of surface current. Therefore, the change

of electron density of Co, Ni and S can be acquired by XANES measurement. We have added more details in the method of the revised manuscript.

2. Could the author give more details on the XANES measurement and the normalization details of the Co and Ni L-edge.

Response : The Co and Ni L-edge XANES spectra were measured at the photoemission end-station at beamline BL10B in the National Synchrotron Radiation Laboratory (NSRL) in Hefei, China. In this experiment, the samples were kept in the total electron yield mode under an ultrahigh vacuum at 5×10^{-10} mbar. The resolving power of the grating was typically $E/\Delta E = 1000$, and the photon flux was 1×10^{10} photons per s. Spectra were collected at energies from 765 to 805 eV for Co L-edge and 835 to 880 eV for Ni L-edge in 0.2 eV energy steps. The NEXAFS raw data were normalized by a procedure consisting of several steps. First, the photon energy was calibrated based on the Au 4f peak of a freshly sputtered gold wafer and then substrate a line to set the pre-edge at zero. Finally, the spectra were normalized to yield an edge-jump to one. We have added the normalization details in the method part.

3. Please label the diffraction peaks in the XRD patterns in Figure 2 and Figure S12.

Response: According to the reviewer's valuable suggestion, the XRD patterns in **Figure 2 and S12** have been well assigned in the revised manuscript.

4. The authors obtained a pretty good catalytic performance on N-NiCo₂S₄ with an overpotential of 41 mV at 10 mA cm⁻². Please compare the HER performance with other metal sulfides reported in literature and make a comparison table in the supplementary information.

Response: Following the suggestions, we have made a table to compare the catalytic performance of N-NiCo₂S₄, with the ever-reported metal sulfides for HER catalysis (Table R1). Based on the comparison, N-NiCo₂S₄ displays the best HER activity. We have added the table in the revised supporting information.

Table R1 | The Summary of the representative metal sulfide-based HER catalysts.

Catalysts	η_{10} (mV)	Tafel slope (mV/ decade)	Electrolyte	Reference
Strained Vacancy-MoS ₂	170	60	acidic	Nature Materials 15 , 48-53 (2016)
Exfoliated Metallic MoS ₂	187	43	acidic	J. Am. Chem. Soc. 135 , 10274-10277 (2013)

NiCo ₂ S ₄	65	84.5	alkaline	Nano Energy 24 , 139-147 (2016)
N-doped and S Vacancy CoS ₂	57	43	acidic	Acs Energy Lett. 2 , 1022-1028 (2017)
NiMo ₃ S ₄	~257	98	alkaline	Angew. Chem. Int. Edit. 128 , 15466-15471 (2016)
P-doped 2H-MoS ₂	130	49	acidic	Adv. Funct. Mater. , 27 (2017)
Cobalt-doped FeS ₂	~100	~46	acidic	J. Am. Chem. Soc. 137 , 1587-1592 (2015)
CoS	~162	93	acidic	J. Am. Chem. Soc. 135 , 17699-17702 (2013)
NiCo ₂ S ₄	210	58.9	alkaline	Adv. Funct. Mater. 26 , 4661-4672 (2016)
WS ₂	~142	70	acidic	Energy Environ. Sci. 7 , 2608-2613 (2014)
CoS ₂	145	51.6	acidic	J. Am. Chem. Soc. 136 , 10053-10061 (2014)
CoPS	48	56	acidic	Nature Materials 14 , 1245-1251 (2015)
Oxygen-Incorporated MoS ₂	~160	55	acidic	J. Am. Chem. Soc. 136 , 17881-17888 (2014)
Exfoliate WS ₂	~221	55	acidic	Nature Materials 12 , 850-855 (2013)
N-NiCo₂S₄	41	37	alkaline	This work

Reviewer #3 (Remarks to the Author):

The manuscript introduces the introduction of NH₃ to the sulfurization of NiCo₂S₄ and the catalytic activity of product (N-NiCo₂S₄) in hydrogen evolution reaction (HER). The performance of N-NiCo₂S₄ is better than that of NiCo₂S₄, and close to that of commercial Pt/C electrocatalyst. The synthesis is convenient, the performance is promising, and the electrochemical characterization is clear. The result would contribute positively to the development of low-cost clean and renewable energy. However, I cannot suggest the acceptance of manuscript because of the following reasons.

Response: We warmly thank the reviewer for the kind comments that the performance of our catalyst is promising and our work would contribute positively to the development of low cost, clean and renewable energy. We also really appreciate the constructive comments the reviewer raised to improve our manuscript. Following

the suggestions, we have made more characterization and experiments to clarify the issues and our point-by-point responses are as follows.

1. The structure characterization of sample needs to be improved. For example, the authors claimed the N was homogeneous in N-NiCo₂S₄, while EDS mapping shows that the signal distributions of Ni, Co, and N cannot overlap each other, implying that the distribution of Ni, Co, and N is inhomogeneous. On the other hand, XRD is hard to reveal small amount of other possible structure in the product. Therefore, EDS mapping with much large magnification (for example, atomic resolution EDS mapping) is necessary to reveal what is the real microstructure of the sample.

Response: We thank the reviewer for the constructive comments. Following the suggestions, we have conducted EDS mapping under both low and high magnification (Figure R10 and R3). In Figure R10, the low-magnification mapping images indicate relative uniform distribution of Ni, Co, S, and N on the single nanowire. Moreover, we also collected the mapping image with much larger magnification shown in Figure R3, which also show nearly uniform element distribution. For the atomic resolution of EDS mapping, it is unfortunately beyond the detection limit of our instrument. Besides, taking the EDS, XPS, XANES, EXAFS and DFT calculation together, it is safe to claim nitrogen is doped in the NiCo₂S₄.

Figure R10 | HAADF-STEM image of a single N-NiCo₂S₄ NW and the element mapping images of Ni, Co, S and N elements in the N-NiCo₂S₄ NW. The scale bar is 100 nm.

Figure R3 | EDS element mapping images of Ni, Co, S and N elements with a higher magnification. The scale bar is 10 nm.

- The authors claimed that the introduction of N improved the intrinsic catalytic activity of N-NiCo₂S₄. The TOFs of N-NiCo₂S₄ and NiCo₂S₄ should be determined to support this claim.

Response: We thank the reviewer for the valuable suggestions. Following the suggestions, we have estimated and compared the per-site turnover frequency (TOF) of the NiCo₂S₄ and N-NiCo₂S₄ for HER catalysis. To calculate the TOF values, we used the previously reported calculation method¹⁻⁴:

$$\text{TOF} = \frac{\text{number of total hydrogen turnovers / cm}^2 \text{ of geometric area}}{\text{number of active sites / cm}^2 \text{ of geometric area}}$$

The total number of hydrogen turnovers (No. of H₂) was obtained by the following equation.

$$\begin{aligned} \text{No. of H}_2 &= \left(j \frac{\text{mA}}{\text{cm}^2} \right) \left(\frac{1 \text{ Cs}^{-1}}{1000 \text{ mA}} \right) \left(\frac{1 \text{ mol e}^-}{96485.3 \text{ C}} \right) \left(\frac{1 \text{ mol H}_2}{2 \text{ mol e}^-} \right) \left(\frac{6.022 \cdot 10^{23} \text{ H}_2 \text{ molecules}}{1 \text{ mol H}_2} \right) \\ &= 3.12 \cdot 10^{15} \frac{\text{H}_2/\text{s}}{\text{cm}^2} \text{ per } \frac{\text{mA}}{\text{cm}^2} \end{aligned}$$

The number of active sites (No. of active sites) was estimated as the number of surface sites (including both Ni, Co and S atoms as the possible active sites). The active sites per real surface area is calculated from the following equation^{3,5}:

$$\text{No. of active sites} = \left(\frac{\text{No. of atoms/unit cell}}{\text{volume/unit cell}} \right)^{\frac{2}{3}}$$

The N-NiCo₂S₄ phase (JCPDS Card No.20-0782), a=b=c=9.329 (from DFT results), contains: 8 Ni, 16 Co, 31 S and 1 N atoms.

$$\text{No. of active sites (N-NiCo}_2\text{S}_4) = \left(\frac{56 \text{ atoms/unit cell}}{811.91 \text{ \AA}^3/\text{unit cell}} \right)^{\frac{2}{3}} = 1.68 \cdot 10^{15} \text{ atoms cm}^{-2};$$

$$\text{No. of active sites (NiCo}_2\text{S}_4) = \left(\frac{56 \text{ atoms/unit cell}}{825.03 \text{ \AA}^3/\text{unit cell}} \right)^{\frac{2}{3}} = 1.66 \cdot 10^{15} \text{ atoms cm}^{-2};$$

Finally, the plot of current density can be converted into a TOF plot according to the following formula²:

$$\text{TOF} = \frac{\left(3.12 \cdot 10^{15} \frac{\text{H}_2/\text{s}}{\text{cm}^2} \text{ per } \frac{\text{mA}}{\text{cm}^2}\right) * |J|}{\text{No. of active sites} * A_{\text{ECSA}}}$$

where the A_{ECSA} is electrochemical surface area (ECSA), which can be estimated using electrochemical double layer capacitance. The calculation equation is as follows².

$$A_{\text{ECSA}} = \frac{\text{specific capacitance}}{40 \mu\text{F cm}^{-2} \text{ per cm}^2_{\text{ECSA}}}$$

Where specific capacitance is C_{dl} ; $40 \mu\text{F cm}^{-2}$ is reported for the calculated ECSA^{2,6}. Figure R11 is the TOF of NiCo_2S_4 and $\text{N-NiCo}_2\text{S}_4$ plot against the potentials. Clearly, the TOF values are significantly enhanced after nitrogen incorporation, suggesting nitrogen doping can intrinsically change catalytic activity of NiCo_2S_4 . We have added this data in the revised manuscript and the calculation details was also provided in the method part.

Figure R11 | TOF against the potentials of NiCo_2S_4 and $\text{N-NiCo}_2\text{S}_4$ for HER catalysis.

3. The XPS spectra should be interpreted in more details. The deconvolution of XPS peaks should be carried out to find out all possible chemical state of Ni, Co, S, N, and O. The sulfide is easy to be oxidized due to air exposure. Actually, strong oxygen peak can be found from the survey XPS spectrum (Figure S4), suggesting the presence of nickel oxide or/and cobalt oxide. The contribution of these oxides to the HER process should be evaluated.

Response: To illuminate the chemical states of Ni, Co, S, N and O, the Hyperfine XPS spectra including Co 2p, Ni 2p, S 2p, N 1s and O 1s of $\text{N-NiCo}_2\text{S}_4$ are shown in Figure R12 a-e. The two core-level peaks of XPS Co 2p at binding energies of 780.0 and 794.7 eV correspond to the Co $2p_{3/2}$ and $2p_{1/2}$ of the typical Co-S bonds⁷⁻⁹, while the peaks at 782.1 and 796.3 eV are the satellite peaks. Similarly, the Ni 2p spectrum can also be deconvoluted into the signals of Ni-S bond and satellite^{9,10}. Meanwhile, no obvious peaks for Co-O (located at 781.1 and 796.1eV) and Ni-O (located at 855.3 and 872.8 eV) can be observed¹¹⁻¹³. For the S 2p in Figure R12c, typical metal-sulfur

binding energies can be observed, while no peak of S-O signal at 168 eV is detected^{7,9}. The XPS N1s shows that the existence of metal-nitrogen bond and surface absorbed amine group. Besides, the O 1s spectrum can be deconvoluted into three peaks which located at 529, 531.3 and 533 eV, which can be ascribed to O-M, O-C and O-H. The O-M signal is very minimum, suggesting the surface oxidation is very light¹⁴⁻¹⁶. The existence of O-C and O-H may be due to the exposure to air for long time, which results in the adsorption of water and some organic impurities during sample operation. We also collected the XPS survey spectrum of the freshly prepared N-NiCo₂S₄ with immediate delivery for sample measurement, which show significantly less oxygen signal, as shown in Figure R12f.

To study the effect of surface oxidation on the HER catalysis, we intentionally heated the N-NiCo₂S₄ in air under different heating temperatures to partially oxidize the surface of N-NiCo₂S₄. With the increase of temperatures, the HER performance continuously decreases (Figure R13), suggesting the surface oxidation has negative effect on the HER catalysis and the observed high performance of N-NiCo₂S₄ in our work is not originated from the surface oxidation. We have added the updated data in the revised supporting information and given a brief discussion in the main text.

Figure R12 | The deconvoluted core-level XPS spectra covering (a) Co 2p, (b) Ni 2p, (c) S 2p, (d) N 1s and (e) O 1s orbitals of N-NiCo₂S₄. (f) XPS survey spectrum of freshly prepared N-NiCo₂S₄.

Figure R13 | The overpotential of N-NiCo₂S₄ NWs after being heated in air at different temperatures at current density of 10 mA cm⁻².

4. The atomic ratio of Ni, Co, S, N and O should be offered. The possibility of defect formation and the consequent influence on the performance should be discussed.

Response: The atomic ratio of Ni, Co, S, N and O is around 1: 2.13: 4.4: 0.67: 1.1, where the introduced O is probably due to the adsorbed oxygen. Following the suggestions, we have conducted two more control experiments to study the effects of defects on the HER catalysis. The defects we consider are surface oxidation and sulfur vacancies. To study the effect of surface oxidation on the HER catalysis, we intentionally heated the N-NiCo₂S₄ in air under different temperatures to partially oxidize the surface of N-NiCo₂S₄. With the increase of temperatures, the HER performance continuously decreases (Figure R13), suggesting the surface oxidation has negative effect on the HER catalysis and the observed high performance of N-NiCo₂S₄ in our work is not originated from the surface oxidation. The defects of sulfur vacancies were also introduced by annealing the NiCo₂S₄ in hydrogen at 300 °C. The overpotential of hydrogen treated NiCo₂S₄ is 81 mV at 10 mA cm⁻² and smaller than NiCo₂S₄ (Figure R14), suggesting the sulfur vacancies have positive effect on the HER catalysis. However, the overall performance of hydrogen treated NiCo₂S₄ is still far less than that of N-NiCo₂S₄, indicating the sulfur vacancy is not the main factor for the HER performance of N-NiCo₂S₄. We have added a brief discussion on the possible effect of defects in the revised manuscript.

Figure R14 | The LSV curves of NiCo₂S₄ (black) and H₂ treated NiCo₂S₄ (red) at 300 °C.

5. The loading amount of N-NiCo₂S₄ should be introduced. Meanwhile, Pt/C should be loaded onto Ni foam with the same loading amount as N-NiCo₂S₄ for the convincing performance comparison. On the other hand, the table that lists the performance of reported electrocatalysts is required.

Response: The loading amount of N-NiCo₂S₄ is 2.3 mg cm⁻². When we load the same amount of Pt/C on nickel foam, the overpotential of Pt/C is 28 mV at 10 mA cm⁻² (Figure R15). The performance of N-NiCo₂S₄ is still close to that of Pt/C. Following the suggestions, we have made a table to compare the catalytic performance of N-NiCo₂S₄ NWs with the ever-reported metal sulfides for HER catalysis (Table R1). Based on the comparison, N-NiCo₂S₄ still displays the best HER activity. We have added them in the revised manuscript and supporting information, and give corresponding discussion.

Figure R15 | Polarization data for 20% Pt/C (2.3 mg cm⁻²) on Ni foam.

Table R1 | Summary of representative metal sulfides-based HER catalyts.

Catalysts	η_{10} (mV)	Tafel slope (mV/ decade)	Electrolyte	Reference
Strained Vacancy-MoS ₂	170	60	acidic	Nature Materials 15 , 48-53 (2016)
Exfoliated Metallic MoS ₂	187	43	acidic	J. Am. Chem. Soc. 135 , 10274-10277 (2013)
NiCo ₂ S ₄	65	84.5	alkaline	Nano Energy 24 , 139-147 (2016)
N-doped and S Vacancy CoS ₂	57	43	acidic	Acs Energy Lett. 2 , 1022-1028 (2017)
NiMo ₃ S ₄	~257	98	alkaline	Angew. Chem. Int. Edit. 128 , 15466-15471 (2016)
P-doped 2H-MoS ₂	130	49	acidic	Adv. Funct. Mater. , 27 (2017)
Cobalt-doped FeS ₂	~100	~46	acidic	J. Am. Chem. Soc.

					137 , 1587-1592 (2015)
CoS	~162	93	acidic		J. Am. Chem. Soc.
					135 , 17699-17702 (2013)
NiCo ₂ S ₄	210	58.9	alkaline		Adv. Funct. Mater.
					26 , 4661-4672 (2016)
WS ₂	~142	70	acidic		Energy Environ. Sci.
					7 , 2608-2613 (2014)
CoS ₂	145	51.6	acidic		J. Am. Chem. Soc.
					136 , 10053-10061 (2014)
CoPS	48	56	acidic		Nature Materials
					14 , 1245-1251 (2015)
Oxygen-Incorporated MoS ₂	~160	55	acidic		J. Am. Chem. Soc.
					136 , 17881-17888 (2014)
Exfoliate WS ₂	~221	55	acidic		Nature Materials
					12 , 850-855 (2013)
N-NiCo₂S₄	41	37	alkaline	This work	

6. The authors claimed that “The CoS₂-terminated NiCo₂S₄ (1 0 0) as a predominant growth surface was modeled...”. However, not experimental data showed the predominant growth surface of NiCo₂S₄ is (100) plane.

Response: We thank the reviewer for the valuable comments. The (1 0 0) plane we chose has two reasons. Our DFT calculation indicates the (100) plane is the most stable facet. In addition, we observed (4 0 0) plane in the HRTEM studies which is parallel to (1 0 0) plane. Furthermore, the diffraction peak of (4 0 0) can also be observed in the XRD pattern of N-NiCo₂S₄. Besides, we also provided the calculation results of (1 0 0) facet with Ni terminated surface (Figure S21), which also show consistent conclusion that N incorporation can suppress the energy barrier for HER catalysis. We have added a brief discussion on it in the revised manuscript.

7. The computation result showed that the d band center of N-NiCo₂S₄ is shifted far from the Fermi level. This should be confirmed by valence band XPS of UPS experiment.

Response: Following the suggestions, we have conducted UPS studies on NiCo₂S₄ and N-NiCo₂S₄. Photoemission spectroscopy was collected at the Photoemission Endstation (BL10B beamline) of the National Synchrotron Radiation Laboratory (NSRL) in Hefei, China. In the measurements of valence band (VB) structure, an excitation of 169.08 eV was utilized. Figure R16 shows the UPS valence band spectra of NiCo₂S₄ and N-NiCo₂S₄. Clearly, the valence band is shifted far from the Fermi level after N doping, which is consistent with our DFT results. We have added the data in the revised supporting information and give corresponding discussion.

Figure R16 | UPS valence band spectra of NiCo₂S₄ (blue) and N-NiCo₂S₄ (red). The binding energy was calibrated and referenced to the E_F of Au foil.

Reference

1. Chen, Z.B. et al. Core-shell MoO₃-MoS₂ nanowires for hydrogen evolution: a functional design for electrocatalytic materials. *Nano Lett.* **11**, 4168-4175 (2011).
2. Zhang, R. et al. Ternary NiCo₂P_x nanowires as pH-universal electrocatalysts for highly efficient hydrogen evolution reaction. *Adv. Mater.* **29**, (2017).
3. Popczun, E.J. et al. Nanostructured nickel phosphide as an electrocatalyst for the hydrogen evolution reaction. *J. Am. Chem. Soc.* **135**, 9267-9270 (2013).
4. Liang H. W. et al. Molecular metal-N_x centres in porous carbon for electrocatalytic hydrogen evolution. *Nat. Commun.* **6** 7992 (2015).
5. Laursen, A.B. et al. Nanocrystalline Ni₅P₄: a hydrogen evolution electrocatalyst of exceptional efficiency in both alkaline and acidic media. *Energy Environ. Sci.* **8**, 1027-1034 (2015).
6. Fang, M. et al. Hierarchical niMo-based 3D electrocatalysts for highly-efficient hydrogen evolution in alkaline conditions. *Nano Energy* **27**, 247-254 (2016).
7. Zhang, J.Y. et al. Activating and optimizing activity of CoS₂ for hydrogen evolution reaction through the synergic effect of N dopants and S vacancies. *Acc Energy Lett.* **2**, 1022-1028 (2017).
8. Liu, G.L. et al. MoS₂ monolayer catalyst doped with isolated Co atoms for the hydrodeoxygenation reaction. *Nat. Chem.* **9**, 810-816 (2017).
9. Zhou, L. et al. Hierarchical CoNi-sulfide nanosheet arrays derived from layered double hydroxides toward efficient hydrazine electrooxidation. *Adv. Mater.* **29** (2017).
10. Sivanantham, A., Ganesan, P. & Shanmugam, S. Hierarchical NiCo₂S₄ nanowire arrays supported on Ni Foam: an efficient and durable bifunctional electrocatalyst for oxygen and hydrogen evolution reactions. *Adv. Funct. Mater.* **26**, 4661-4672 (2016).
11. Gao, X.H. et al. Hierarchical NiCo₂O₄ hollow microcuboids as bifunctional electrocatalysts for overall water-splitting. *Angew. Chem. Int. Ed.* **55**, 6290-6294

- (2016).
12. Yu, X.X. et al. Direct growth of porous crystalline NiCo₂O₄ nanowire arrays on a conductive electrode for high-performance electrocatalytic water oxidation. *J. Mater. Chem. A* **2**, 20823-20831 (2014).
 13. Chen, R., Wang, H.Y., Miao, J.W., Yang, H.B. & Liu, B. A flexible high-performance oxygen evolution electrode with three-dimensional NiCo₂O₄ core-shell nanowires. *Nano Energy* **11**, 333-340 (2015).
 14. Salah, N. et al. Flow controlled fabrication of N doped ZnO thin films and estimation of their performance for sunlight photocatalytic decontamination of water. *Chem. Eng. J.* **291**, 115-127 (2016).
 15. Lee, C. & Lee, S.Y. Mussel-inspired bolaamphiphile sticky self-assemblies for the preparation of magnetic nanoparticles. *Colloid Surface B.* **127**, 89-95 (2015).
 16. Yin J. et al. Oxygen vacancies dominated NiS₂/CoS₂ interface porous nanowires for portable Zn–Air batteries driven water splitting devices. *Adv. Mater.* **29**, (2017).

Reviewers' Comments:

Reviewer #1 (Remarks to the Author):

Dear Authors

The authors have answered my questions and concerns. I believe that paper is now stronger and a good fit for nature communications. The only outstanding issue is the influence of d band position and H binding. However, this is in my opinion also outside the scope of the paper since it related to the coupling between electronic structure and reactivity of sulfide surfaces which is in general not very well understood. So even though I personally believe that the analysis and conclusion of the authors are wrong, I also recognize that this an open research question.

I recommend publication as is

Reviewer #2 (Remarks to the Author):

Authors addressed all the questions carefully and the revised manuscript is satisfactory. I recommend it for publication at the present form

Reviewer #3 (Remarks to the Author):

The questions I concerned have been answered. I suggest the acceptance of manuscript.

Response letter

We sincerely thank the referees for their careful review and valuable comments, which certainly help improve our manuscript.

Reviewer #1 (Remarks to the Author):

Dear Authors

The authors have answered my questions and concerns. I believe that paper is now stronger and a good fit for *Nature Communications*. The only outstanding issue is the influence of d band position and H binding. However, this is in my opinion also outside the scope of the paper since it related to the coupling between electronic structure and reactivity of sulfide surfaces which is in general not very well understood. So even though I personally believe that the analysis and conclusion of the authors are wrong, I also recognize that this an open research question.

I recommend publication as is.

Response: We warmly thank the reviewer for recommending publication as it in *Nature Communications*. Yes, we agree with the fact that the influence of d band position on H binding is currently an open research question, and we also added a brief discussion on it in the revised manuscript.

Reviewer #2 (Remarks to the Author):

Authors addressed all the questions carefully and the revised manuscript is satisfactory. I recommend it for publication at the present form

Response: We warmly thank the reviewer for recommending our work for publication in *Nature Communications*.

Reviewer #3 (Remarks to the Author):

The questions I concerned have been answered. I suggest the acceptance of manuscript.

Response: We warmly thank the reviewer for recommending our work for publication in *Nature Communications*.